# Learning the Neighborhood: Contrast-Free Multimodal Self-Supervised Molecular Graph Pretraining

Boshra Ariguib [1]   Mathias Niepert [1]   Andrei Manolache [1,2]

## Abstract

High-quality molecular representations are essential for property prediction and molecular design, yet large labeled datasets remain scarce. While self-supervised pretraining on molecular graphs has shown promise, many existing approaches either depend on hand-crafted augmentations or complex generative objectives, and often rely solely on 2D topology, leaving valuable 3D structural information underutilized. To address this gap, we introduce C-FREE (*Contrast-Free Representation learning on Ego-nets*), a simple framework that integrates 2D graphs with ensembles of 3D conformers. C-FREE learns molecular representations by predicting subgraph embeddings from their complementary neighborhoods in the latent space, using fixed-radius ego-nets as modeling units across different conformers. This design allows us to integrate both geometric and topological information within a hybrid Graph Neural Network (GNN)-Transformer backbone, without negatives, positional encodings, or expensive pre-processing. Pretraining on the GEOM dataset, which provides rich 3D conformational diversity, C-FREE achieves state-of-the-art results on MoleculeNet, surpassing contrastive, generative, and other multimodal self-supervised methods. Fine-tuning across datasets with diverse sizes and molecule types further demonstrates that pretraining transfers effectively to new chemical domains, highlighting the importance of 3D-informed molecular representations. Our code is available at https://github.com/ariguiba/C-FREE.

## 1. Introduction

High-quality molecular representations are critical for predicting properties, interpreting chemical behavior, and accelerating compound discovery (Wigh et al., 2022; Elton et al., 2019). Many existing approaches, however, rely on a single modality, such as SMILES strings (Hirohara et al., 2018; Wang et al., 2019), 2D graph structures (Gilmer et al., 2017; Kipf & Welling, 2017; Xu et al., 2019), or 3D conformations (Schütt et al., 2017; Gasteiger et al., 2021). While effective, each of these methods captures only part of the molecular information and overlooks complementary aspects available in other modalities (Liu et al., 2023b). Beyond modality limitations, these models often require large, curated datasets, which restricts their use in low-data settings.

Because such curated datasets are often unavailable, especially in low-data regimes, self-supervised learning (SSL) provides a promising alternative to fully supervised training. Recent advances in vision and language modeling (Devlin et al., 2019; Chen et al., 2020; Grill et al., 2020a; Caron et al., 2021; He et al., 2022; Dawid & LeCun, 2024) have motivated similar methods for molecular graphs, especially approaches that aim to combine structural and geometric information. While these approaches have advanced molecular representation learning, each comes with trade-offs: contrastive methods hinge on carefully chosen negative samples (You et al., 2021b;a), generative methods often require discrete reconstruction in the input graph space with graph tokenization (Hu et al., 2020b), and latent-predictive methods can depend on augmentations or expensive procedures such as clustering (Skenderi et al., 2025). These challenges motivate simpler predictive frameworks that combine 2D topology and 3D conformations without relying on negatives or input-space reconstruction, and that work reliably across both high- and low-data settings.

**Current work.** Our self-supervised framework C-FREE (Contrast-Free Representation learning on Ego-nets) adopts a predictive learning strategy with subgraphs as the basic modeling unit. It is motivated by three goals: (i) *avoiding computationally intensive or ambiguous design choices*, including expensive augmentations, heavy subgraph-construction procedures, and complex negative-

[1]Computer Science Department, University of Stuttgart, Germany [2]Bitdefender, Romania. Correspondence to: Boshra Ariguib <ariguiba@studi.informatik.uni-stuttgart.de>.

*Proceedings of the 43rd International Conference on Machine Learning*, Seoul, South Korea. PMLR 306, 2026.

sampling schemes. For example, clustering-based subgraph algorithms such as METIS (Skenderi et al., 2025) can be costly, and defining suitable augmentations or negatives is often non-trivial, since molecules with nearly identical structures (e.g., chiral isomers) may still have very different properties; (ii) *leveraging the success of subgraph-based methods in 2D graph supervised learning* (Bevilacqua et al., 2022; Wollschläger et al., 2024), which suggest that aggregating information from substructures can yield richer graph-level representations, and (iii) *harnessing the benefits of multimodal architectures in the supervised setting* (Zhu et al., 2024; Nguyen et al., 2024; Manolache et al., 2024). Since many molecular properties depend on multiple conformations and their probabilities (Cao et al., 2022), using multiple high-probability conformers alongside 2D topology helps capture this variability and improves predictive performance. Building on JEPA (Assran et al., 2023) and Equivariant Subgraph Aggregation Networks (ESAN) (Bevilacqua et al., 2022), our method segments graphs into disjoint subgraphs, similar to image patches or language tokens, and learns to align each subgraph with its context in latent space. Unlike GraphJEPA (Skenderi et al., 2025) and I-JEPA (Assran et al., 2023), it avoids positional encodings, hierarchical objectives, and costly clustering, and instead leverages the inductive bias of 2D and 3D encoders together with subgraph-based pretraining to learn rich embeddings. Our contributions are as follows:

1. **A new multimodal pretraining task for molecular graphs.** We introduce a broadly applicable predictive objective based on $k$-EgoNet subgraphs, avoiding costly hand-crafted augmentations and utilizing both 2D and 3D views of the molecule.

2. **Robust performance in both multimodal and 2D-only settings.** Our framework leverages 2D topology together with multiple 3D conformations when available, but also performs strongly in purely 2D settings where conformers are absent.

3. **A simple and effective training scheme.** We adopt non-contrastive predictive learning, avoiding the pretrain/fine-tune mismatch and removing the need for negative samples or augmentations. Moreover, when fine-tuning, our framework simulates ESAN (Bevilacqua et al., 2022) and is provably more expressive than 1-WL (Weisfeiler & Leman, 1968)

4. **State-of-the-art results.** Our approach matches or surpasses other self-supervised models under both linear-probe evaluation, where the backbone is frozen, and full fine-tuning. It achieves the best average performance on MoleculeNet (Wu et al., 2018) and shows strong transfer to novel multimodal molecular benchmarks such as MARCEL (Zhu et al., 2024).

## 2. Related Work

Existing approaches to graph self-supervised learning can be grouped into three main categories: *contrastive learning*, *generative pretraining*, and *latent representation learning*. Each of these has also been extended to molecular graphs, with varying degrees of multimodal integration.

*Contrastive learning* aligns representations of similar instances while pushing apart dissimilar ones and has become central to graph representation learning. GraphCL (You et al., 2021b) and JOAO (You et al., 2021a) pioneered this idea through graph augmentations, while InfoGraph (Sun et al., 2020) maximized mutual information across views. Extensions to molecules incorporate 3D information: Graph-MVP (Liu et al., 2022a) aligns 2D topology and 3D conformations with generative objectives, MoleculeSDE (Liu et al., 2023a) introduces symmetry-aware stochastic differential equations, and 3D InfoMax (Stärk et al., 2022) encodes 3D from 2D via mutual information. While effective (Wang et al., 2023a), these methods depend on negative samples and large batches (You et al., 2021b), a limitation exacerbated by irregular graph structures.

*Generative pretraining* forms the second category of self-supervised learning, where models reconstructs masked or missing components of a graph from its surrounding context. Early works such as AttrMask (Hu et al., 2020a), ContextPred (Hu et al., 2020a), and EdgePred (Hamilton et al., 2017) predict attributes or edges, while GPT-GNN (Hu et al., 2020b) and GROVER (Rong et al., 2020) extend to autoregressive reconstruction and chemically informed motif prediction. More recent methods integrate multimodal and geometric signals: unified cross-modal generation of 2D/3D (Zhu et al., 2022), modality integration via Mole-Blend (Yu et al., 2023), and geometry-aware prediction in 3D PGT (Wang et al., 2023b). Despite their promise, generative methods must reconstruct both discrete graph structure and continuous features, with autoregressive variants further complicated by the lack of natural node ordering.

*Latent representation learning* forms the third category of self-supervised methods. Instead of reconstructing raw graph structures or features, these approaches predict target embeddings directly in latent space, yielding compact, denoised, and often across multimodal representations. BGRL (Thakoor et al., 2023) employs a bootstrapped online–target encoder scheme under augmentations, while LaGraph (Xie et al., 2022) frames the task as latent graph prediction, optimizing an upper bound with context-aware regularization on masked nodes. While latent prediction methods avoid the costly generation of negatives, they remain sensitive to augmentation quality and model update stability, and are prone to representation collapse (Assran et al., 2023; Grill et al., 2020b).

Within latent representation learning, GraphJEPA (Skenderi

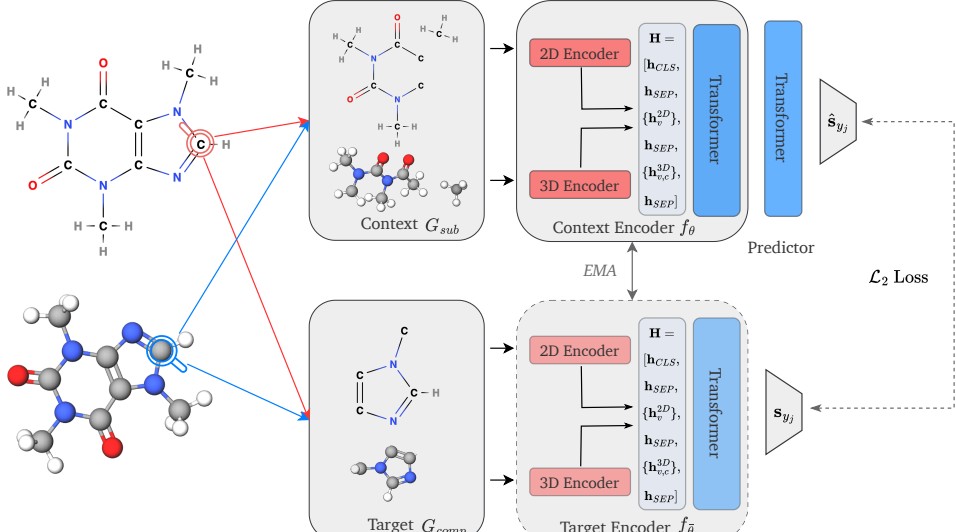

*Figure 1.* From each molecular graph, we sample a random node and extract its k-EgoNet (Bevilacqua et al., 2022) with $k \in \{3, 4\}$ to form complementary context and target subgraphs. Both 2D and 3D views are encoded with a GINE and a PaiNN, concatenated, and passed through a transformer; the context embedding is further processed by a predictor to estimate the target. Training minimizes the mean squared $\mathcal{L}_2$ loss between predicted and encoded targets, with the target encoder updated as an exponential moving average (EMA) of the context encoder (Grill et al., 2020b; Assran et al., 2023). Though we show one 3D conformation, we use three in practice. During fine-tuning, the target encoder serves as the pretrained backbone with task-specific heads added. See Appendix Figure 5 for details.

et al., 2025) extends the Joint Embedding Predictive Architecture (JEPA) (Assran et al., 2023) to graphs by masking METIS-generated clusters and predicting them as patch-like substructures, while also encoding hierarchical information via hyperbolic subgraph coordinates. While effective, this approach incurs significant computational cost and depends on auxiliary components —such as clustering, hierarchical encodings, positional embeddings— that add complexity without being clearly essential for representation learning.

Another direction explores large-scale supervised pretraining on massive labeled molecular datasets, aiming to transfer knowledge to downstream tasks (Beaini et al., 2024). While effective in some cases, this approach still depends on labeled data and may be domain-specific, since source and target distributions can differ. In contrast, self-supervised methods avoid this reliance on labels and can transfer more flexibly. Importantly, the two strategies are complementary: self-supervised pretraining can provide strong initializations that are later fine-tuned on labeled data.

To the best of our knowledge, our work is the first non-contrastive, non-generative predictive framework for multi-modal molecular representation learning. We now turn to its design.

## 3. Contrast-Free Multimodal Self-Supervised Pretraining

In the following, we outline our proposed training pipeline, illustrated in Figure 1. Unlike most generative methods (Hu

et al., 2020a; Hamilton et al., 2017), we apply our training objective fully in the latent space, without reconstructing the original features of the masked components.

The core principle of our approach is to learn representations by aligning the embedding of a target view with that of a related context view. Specifically, we represent each molecule as a 2D graph $G = (V, E)$, where $V$ is the set of nodes (atoms) and $E$ the set of edges (covalent bonds). For each atom $v \in V$, we include 3D coordinates $r_v \in \mathbb{R}^3$, taken from multiple conformers of the molecule. Using these graph and geometric features, we construct complementary context–target views that serve as inputs for our contrast-free pretraining scheme. Each 2D and 3D view is encoded independently, and the resulting embeddings are concatenated into a single multimodal sequence processed by a transformer (Vaswani et al., 2017). We then align the embedding of the target subgraph with that of its associated context subgraph. This design is loosely inspired by ESAN (Bevilacqua et al., 2022), but adopts a simplified variant: we use fixed-radius ego-nets to obtain complementary views during pretraining, and at fine-tuning we evaluate both linear probing on whole-graph embeddings and an aggregation of subgraph embeddings using DeepSets (Zaheer et al., 2017).

**Context–Target View Generation.** We generate complementary 2D views by sampling $k$-EgoNets, where the $k$-hop neighborhood of a node defines one subgraph and the remaining nodes and edges define its complement (see Fig-

ure 3 in the Appendix). Specifically, we define the k-ego-net of a node as its k-hop neighbourhood with the induced connectivity. Building on this, we define complementary subgraphs as follows: the context consists of the k-EgoNet of an anchor node, while the target comprises the remaining edges not included in the context. To guarantee full edge coverage, edges on the boundary between the two subgraphs are assigned exclusively to one subgraph; boundary nodes, however, are duplicated and appear in both, ensuring the two subgraphs are edge-disjoint. The 3D coordinates are added to generate the corresponding 3D views for all the conformers. Either view can serve as the target while the other acts as context, and their roles are alternated during training to avoid prediction bias. We adopt fixed-radius neighborhoods with $k \in \{3, 4\}$, analogous to fixed-size patches in vision-based methods (Assran et al., 2023). Although graphs vary in size and structure, this ensures that each subgraph captures a comparable amount of local information. To further diversify training, we sample multiple nodes $v_1, v_2, \ldots, v_n$ per molecule and construct their corresponding k-EgoNets $E(v_1), E(v_2), \ldots, E(v_n)$, yielding multiple complementary context–target pairs without increasing dataset size.

**Context Encoder.** We aim to learn subgraph representations that generalize to whole-molecule embeddings. Following the architecture proposed in (Manolache et al., 2024), we use a message-passing neural network (MPNN) with GINE (Hu et al., 2020a; Xu et al., 2019) as the 2D encoder, and experiment with both PaiNN (Schütt et al., 2021) and SchNet (Schütt et al., 2017) as the 3D encoder used to process multiple conformers. We restrict our framework to 2D and 3D modalities; we discuss the exclusion of 1D representations such as SMILES in Appendix A.4.

From GINE, we obtain node-level embeddings $\{\mathbf{h}_v^{2D}\}$ for all atoms in the subgraph by averaging their intermediate representations across layers. From the 3D encoder, we extract node-level embeddings $\{\mathbf{h}_{v,c}^{3D}\}$ for each conformer $c$, preserving per-atom detail across conformations. To build the multimodal sequence, we prepend a learnable classification token $\mathbf{h}_{CLS}$ and insert a learnable separation token $\mathbf{h}_{SEP}$ between the 2D and 3D components, resulting in the following multimodal sequence:

$$\mathbf{H} = [\mathbf{h}_{CLS}, \mathbf{h}_{SEP}, \{\mathbf{h}_v^{2D}\}, \mathbf{h}_{SEP}, \{\mathbf{h}_{v,c}^{3D}\}, \mathbf{h}_{SEP}]$$

To distinguish between modalities, we add learnable modality embeddings that mark whether a token comes from the 2D or 3D graph. The full sequence is then passed through a Transformer with multiple self-attention layers to capture global dependencies both within and across modalities.

**Predictor Network.** The predictor takes as input the full

multimodal token sequence, after it has been processed by the context encoder. The predictor is a lightweight transformer followed by an MLP, which maps the context embedding to the representation of the complementary subgraph. Since the upstream modality encoders already capture spatial and relational information, we do not add explicit positional encodings, unlike image-based JEPA (Assran et al., 2023) and 2D GraphJEPA (Skenderi et al., 2025).

**Target Encoder.** The target subgraph $f_{\bar{\theta}}$ is encoded by a separate instance of the context encoder. Maintaining two distinct networks stabilizes training and mitigates representation collapse, a strategy widely adopted in self-predictive frameworks such as BYOL (Grill et al., 2020b), I-JEPA (Assran et al., 2023), and BGRL (Thakoor et al., 2023). The target encoder's weights are updated via an exponential moving average (EMA) of the context encoder's parameters:

$$\bar{\theta}^{(t)} = \tau \, \bar{\theta}^{(t-1)} + (1 - \tau) \, \theta^{(t)},$$

where $\bar{\theta}^{(t)}$ are the exponentially moving averaged parameters at step $t$, $\theta^{(t)}$ are the current context encoder parameters, and $\tau \in [0, 1]$ is the decay rate controlling the contribution of past parameters.

**Pretraining task.** Each subgraph is represented by a single multimodal embedding, taken from the final classification token embedding $\mathbf{h}_{CLS}^{out}$. For the *context* subgraph, we feed the entire multimodal token sequence from the context encoder into the predictor transformer and take its output CLS token as the predicted embedding. For the *target* subgraph, we use the CLS token directly from the target encoder. The self-supervised objective minimizes the mean squared $\mathcal{L}_2$ distance between the predicted context embedding and the target embedding:

$$\frac{1}{M} \sum_{i=1}^{M} \sum_{j=1}^{N_G} \left\| \hat{\mathbf{s}}_{y_j} - \mathbf{s}_{y_j} \right\|^2,$$

where $\hat{\mathbf{s}}_{y_j}$ and $\mathbf{s}_{y_j}$ denote the predicted and target subgraph embeddings, $M$ is the batch size, and $N_G$ the number of sampled views (ego-nets and their complements). All views are treated as separate instances when computing the loss.

**Fine-tuning.** When fine-tuning, we use the target encoder as our pretrained backbone to generate graph embeddings, and add lightweight task-specific heads. We consider two types of task heads: (i) linear probing on whole molecule embeddings with a single linear layer (C-FREE$_{MOL}$) to evaluate representation quality, and (ii) aggregating $k$-EgoNet subgraph embeddings with DeepSets (Zaheer et al., 2017) (C-FREE$_{SUB}$), showing that subgraph pretraining transfers both to whole-molecule prediction and to ESAN-style fine-tuning schemes. We find that C-FREE$_{SUB}$ is especially beneficial in the 2D-only setting, while for multimodal inputs

both heads perform similarly, likely because 3D information compensates for the lower expressiveness. Nevertheless, downstream convergence is faster with DeepSets, suggesting advantages in aligning pretraining and fine-tuning.

### 3.1. Invariance and Expressiveness

We make two theoretical observations. First, with the DeepSets head, C-FREE$_{SUB}$ simulates ESAN (Bevilacqua et al., 2022) and is more expressive than 1-WL (Weisfeiler & Leman, 1968). An informal statement is given below; the full theorem and proof are in Appendix Section A.1.

**(Informal) Lemma 1.** *Under the assumptions from Theorem 2 of (Bevilacqua et al., 2022), C-FREE with a DeepSets task head is as expressive as ESAN, hence it is strictly more expressive than the* 1-*WL algorithm (Weisfeiler & Leman, 1968).*

Second, C-FREE preserves the invariances of its modality encoders. Prior work has shown that architectures of this form inherit invariance from their encoders (Manolache et al., 2024), and the same holds for our framework. For completeness, we include the lemma in Appendix Section A.1. As empirical validation, we also provide an experiment on the EXP dataset in the Appendix A.2.

## 4. Empirical Evaluation

We evaluate our framework through four complementary sets of experiments:

(i) **Frozen backbone evaluation.** We assess representation quality by freezing the backbone and training linear probes (Section 4.1). On MoleculeNet (Wu et al., 2018), C-FREE outperforms contrastive and non-contrastive baselines and remains effective even with 2D-only inputs (Table 1).

(ii) **Full fine-tuning.** We further evaluate end-to-end adaptability by updating the full backbone with task-specific heads. On MoleculeNet, this setup tests how well pretraining transfers to dataset-specific classification tasks (Section 4.2.1). On QM9, we compare against larger baseline models and evaluate chemically informed subgraph pretraining (Section 4.2.2). On Kraken, we find that pretraining improves downstream regression performance over random initialization (Table 5). Finally, on the larger Drugs-75K dataset, we study label efficiency by fine-tuning on progressively larger subsets of labeled data (Section 4.2.3).

(iii) **Ablations.** We conduct four ablations: (i) removing each modality to assess its contribution (Section 4.3), (ii) removing the predictor network to evaluate its impact on training stability (Section 4.4), (iii) varying

the EMA decay rate to assess representation quality (Section 4.5), and (iv) using single k-EgoNet subgraph types to evaluate the effect of the radius parameter $k$ (Section 4.6).

(iv) **Theory alignment.** We verify whether the empirical expressiveness aligns with the theoretical result from Lemma 1 (Section A.2).

Implementation details for pretraining and evaluation are provided in Section A.7 in the Appendix.

### 4.1. Comparison with Frozen Backbones

For the first experiments, we compare our framework against state-of-the-art contrastive and non-contrastive self-supervised methods on molecular property classification tasks. Following Wang et al. (2023a), we pretrain two backbones on 0.33M molecules from GEOM (Axelrod & Gomez-Bombarelli, 2022) and evaluate them on MoleculeNet (Wu et al., 2018). One backbone has 4M parameters and uses 2D inputs, while the other has 9.1M parameters and incorporates both 2D graphs and 3D conformer ensembles available in GEOM, with three additional conformers generated using RDKit (Landrum, 2016) at fine-tuning. This setup enables fair comparison to 2D-only baselines while also testing the benefit of 3D information. Performance is reported as mean ROC-AUC over three scaffold splits, with frozen checkpoints chosen by best self-supervised loss and linear probes selected by downstream validation loss. We evaluate two strategies: linear probing on whole-graph embeddings and on subgraph embeddings aggregated with DeepSets (Zaheer et al., 2017). We also run the same experiment on molecular property regression using the Kraken dataset (Gensch et al., 2022) (see Appendix Section A.12).

As shown in Table 1, our framework achieves the best average performance, outperforming baselines on 6 of 8 tasks. In the 2D-only setting, DeepSets aggregation provides clear gains and yields the best average result, with further improvements from adding 3D information. We also see strong gains on multi-task datasets such as Tox21, ToxCast, and MUV, suggesting that our model captures more generalizable features. While DeepSets helps in 2D-only, its effect is minimal in the multimodal case, likely because 3D inputs already encode rich structural detail.

### 4.2. Full Fine-tuning

Beyond evaluating frozen representations, we next assess the adaptability of our pretrained models through full end-to-end fine-tuning. In this setting, the entire backbone is updated jointly with a task-specific head, enabling us to test how pretraining improves convergence and downstream performance. We consider two scenarios: (i) property classification on MoleculeNet, where we compare against both contrastive, non-contrastive and multimodal self-supervised

*Table 1.* Performance on MoleculeNet (Wu et al., 2018) with 2D-only frozen backbones. **Non-CL** denotes non-contrastive and **CL** contrastive methods. We report **C-FREE$_{2D}$** (2D-only) and **C-FREE$_{MM}$** (multimodal), each with linear probing on whole-molecule embeddings (**MOL**) or on subgraphs using subgraph aggregation with DeepSets (Zaheer et al., 2017) (**SUB**). Metric: ROC-AUC (↑). Red marks the best model and Blue the second best. **C-FREE** ranks first or second on 6 of 8 datasets, with **MM-MOL** best overall, while even the 2D-only variants of C-FREE outperform all baselines on average. The full table from Wang et al. (2023a) can be found in the Appendix Section A.13

| | | BBBP (↑) | Tox21 (↑) | ToxCast (↑) | Sider (↑) | ClinTox (↑) | MUV (↑) | HIV (↑) | Bace (↑) | Avg (↑) |
|---|---|---|---|---|---|---|---|---|---|---|
| | | \multicolumn MoleculeNet Datasets (Linear Probe) | | | | | | | | |
| | Random init. | $50.7_{\pm2.5}$ | $64.9_{\pm0.5}$ | $53.2_{\pm0.3}$ | $53.2_{\pm1.1}$ | $63.1_{\pm2.3}$ | $62.1_{\pm1.3}$ | $66.1_{\pm0.7}$ | $63.4_{\pm1.8}$ | 59.60 |
| CL | InfoGraph | $65.9_{\pm0.6}$ | $65.8_{\pm0.7}$ | $54.6_{\pm0.1}$ | $57.2_{\pm1.0}$ | $61.4_{\pm4.8}$ | $63.9_{\pm1.9}$ | $71.4_{\pm0.6}$ | $67.4_{\pm4.9}$ | 63.44 |
| | GROVER | $67.0_{\pm0.3}$ | $63.9_{\pm0.3}$ | $53.6_{\pm0.4}$ | $59.9_{\pm1.7}$ | $65.0_{\pm6.4}$ | $62.7_{\pm1.4}$ | $67.8_{\pm1.0}$ | $69.0_{\pm4.7}$ | 63.62 |
| | GraphCL | $64.7_{\pm1.7}$ | $69.1_{\pm0.5}$ | $56.2_{\pm0.2}$ | $59.5_{\pm0.9}$ | $60.8_{\pm3.0}$ | $60.6_{\pm1.8}$ | $72.5_{\pm1.4}$ | $77.0_{\pm1.7}$ | 65.04 |
| Non-CL | EdgePred | $54.2_{\pm1.0}$ | $66.2_{\pm0.2}$ | $54.4_{\pm0.1}$ | $56.1_{\pm0.1}$ | $65.4_{\pm5.0}$ | $59.5_{\pm0.9}$ | $73.6_{\pm0.4}$ | $71.4_{\pm1.2}$ | 62.59 |
| | AttrMask | $62.7_{\pm2.7}$ | $65.7_{\pm0.8}$ | $56.1_{\pm0.2}$ | $58.3_{\pm1.5}$ | $61.9_{\pm6.4}$ | $60.9_{\pm1.8}$ | $65.5_{\pm1.4}$ | $64.8_{\pm2.6}$ | 61.99 |
| | ContextPred | $55.5_{\pm2.0}$ | $67.9_{\pm0.7}$ | $54.0_{\pm0.3}$ | $57.1_{\pm0.5}$ | $67.4_{\pm4.3}$ | $60.5_{\pm0.9}$ | $66.2_{\pm1.5}$ | $54.4_{\pm3.2}$ | 60.36 |
| | C-FREE$_{2D\text{-}MOL}$ | $60.5_{\pm1.7}$ | $76.1_{\pm0.2}$ | $62.7_{\pm0.4}$ | $59.0_{\pm0.6}$ | $62.7_{\pm1.0}$ | $67.6_{\pm0.5}$ | $68.7_{\pm0.4}$ | $75.8_{\pm0.9}$ | 66.63 |
| | C-FREE$_{2D\text{-}SUB}$ | $64.2_{\pm3.8}$ | $76.7_{\pm0.6}$ | $63.9_{\pm0.3}$ | $58.0_{\pm0.5}$ | $71.4_{\pm3.7}$ | $64.6_{\pm3.1}$ | $65.5_{\pm0.6}$ | $73.9_{\pm0.7}$ | 67.27 |
| | C-FREE$_{MM\text{-}MOL}$ | $69.8_{\pm2.6}$ | $79.9_{\pm1.1}$ | $65.8_{\pm0.7}$ | $58.5_{\pm2.5}$ | $69.9_{\pm1.9}$ | $76.6_{\pm2.8}$ | $72.8_{\pm0.7}$ | $75.3_{\pm1.1}$ | 71.07 |
| | C-FREE$_{MM\text{-}SUB}$ | $73.8_{\pm2.1}$ | $76.7_{\pm0.7}$ | $66.8_{\pm0.2}$ | $56.4_{\pm1.5}$ | $75.7_{\pm2.2}$ | $70.6_{\pm1.0}$ | $71.9_{\pm1.5}$ | $75.5_{\pm1.9}$ | 70.92 |

*Table 2.* Performance on MoleculeNet (Wu et al., 2018) with full end-to-end fine-tuning. For this experiment, the downstream model receives the entire molecule as input, and we limit evaluation to the multimodal variant. **Non-CL** denotes non-contrastive, **CL** contrastive, and **Multi** multimodal methods. Our model is reported as **C-FREE$_{MM\text{-}MOL}$**, with the multimodal variant using both modalities. The evaluation metric is ROC-AUC (↑). Red highlights the best results and Blue the second best. **C-FREE** achieves the best results on 4 out of 8 datasets and ranks first overall, outperforming both multimodal baselines.

| | | BBBP (↑) | Tox21 (↑) | ToxCast (↑) | Sider (↑) | ClinTox (↑) | MUV (↑) | HIV (↑) | Bace (↑) | Avg (↑) |
|---|---|---|---|---|---|---|---|---|---|---|
| | | \multicolumn MoleculeNet Datasets (Full Fine-Tuning) | | | | | | | | |
| CL | InfoGraph | $67.5_{\pm0.1}$ | $73.2_{\pm0.4}$ | $63.7_{\pm0.5}$ | $59.9_{\pm0.3}$ | $76.5_{\pm1.0}$ | $74.1_{\pm0.7}$ | $75.1_{\pm0.9}$ | $77.8_{\pm0.8}$ | 70.98 |
| | GROVER | $70.0_{\pm0.1}$ | $74.3_{\pm0.1}$ | $65.4_{\pm0.4}$ | $64.8_{\pm0.6}$ | $81.2_{\pm3.0}$ | $67.3_{\pm1.8}$ | $62.5_{\pm0.9}$ | $82.6_{\pm0.7}$ | 71.01 |
| | GraphCL | $69.7_{\pm0.6}$ | $73.9_{\pm0.6}$ | $62.4_{\pm0.5}$ | $60.5_{\pm0.8}$ | $76.0_{\pm2.6}$ | $69.8_{\pm2.6}$ | $78.5_{\pm1.2}$ | $75.4_{\pm1.4}$ | 70.78 |
| | MolCLR | $66.6_{\pm1.8}$ | $73.0_{\pm0.1}$ | $62.9_{\pm0.3}$ | $57.5_{\pm1.7}$ | $86.1_{\pm0.9}$ | $72.5_{\pm2.3}$ | $76.2_{\pm1.5}$ | $71.5_{\pm3.1}$ | 70.79 |
| | GraphLoG | $72.5_{\pm0.8}$ | $75.7_{\pm0.5}$ | $63.5_{\pm0.7}$ | $61.2_{\pm1.1}$ | $76.7_{\pm3.3}$ | $76.0_{\pm1.1}$ | $77.8_{\pm0.8}$ | $83.5_{\pm1.2}$ | 73.40 |
| Non-CL | AttrMask | $65.0_{\pm2.3}$ | $74.8_{\pm0.2}$ | $62.9_{\pm0.1}$ | $61.2_{\pm0.1}$ | $87.7_{\pm1.1}$ | $73.4_{\pm2.0}$ | $76.8_{\pm0.5}$ | $79.7_{\pm0.3}$ | 72.68 |
| | ContextPred | $65.7_{\pm0.6}$ | $74.2_{\pm0.0}$ | $62.5_{\pm0.3}$ | $62.2_{\pm0.5}$ | $77.2_{\pm0.8}$ | $75.3_{\pm1.5}$ | $77.1_{\pm0.8}$ | $76.0_{\pm2.0}$ | 71.28 |
| | GraphMAE | $72.0_{\pm0.6}$ | $75.5_{\pm0.6}$ | $64.1_{\pm0.3}$ | $60.3_{\pm1.1}$ | $82.3_{\pm1.2}$ | $76.3_{\pm2.4}$ | $77.2_{\pm1.0}$ | $83.1_{\pm0.9}$ | 73.85 |
| | MGSSL | $69.7_{\pm0.9}$ | $76.5_{\pm0.3}$ | $64.1_{\pm0.7}$ | $61.8_{\pm0.8}$ | $80.7_{\pm2.1}$ | $78.7_{\pm1.5}$ | $78.8_{\pm1.2}$ | $79.1_{\pm0.9}$ | 73.70 |
| | Mole-BERT | $71.9_{\pm1.6}$ | $76.8_{\pm0.5}$ | $64.3_{\pm0.2}$ | $62.8_{\pm1.1}$ | $78.9_{\pm3.0}$ | $78.6_{\pm1.8}$ | $78.2_{\pm0.8}$ | $80.8_{\pm1.4}$ | 74.04 |
| Multi | GraphMVP | $68.5_{\pm0.2}$ | $74.5_{\pm0.4}$ | $62.7_{\pm0.1}$ | $62.3_{\pm1.6}$ | $79.0_{\pm2.5}$ | $75.0_{\pm1.4}$ | $74.8_{\pm1.4}$ | $76.8_{\pm1.1}$ | 71.69 |
| | 3D InfoMax | $69.1_{\pm1.0}$ | $74.5_{\pm0.7}$ | $64.4_{\pm0.8}$ | $60.6_{\pm0.7}$ | $79.9_{\pm3.4}$ | $74.4_{\pm2.4}$ | $76.1_{\pm1.3}$ | $79.7_{\pm1.5}$ | 72.34 |
| | MoleculeSDE | $71.8_{\pm0.7}$ | $76.8_{\pm0.3}$ | $65.0_{\pm0.2}$ | $60.8_{\pm0.3}$ | $87.0_{\pm0.5}$ | $80.9_{\pm0.3}$ | $78.8_{\pm0.9}$ | $79.5_{\pm2.1}$ | 75.07 |
| | MoleBlend | $73.0_{\pm0.8}$ | $77.8_{\pm0.4}$ | $66.1_{\pm0.0}$ | $64.9_{\pm0.3}$ | $87.6_{\pm0.7}$ | $77.2_{\pm2.3}$ | $79.0_{\pm0.6}$ | $83.7_{\pm1.4}$ | 76.16 |
| | MolFM | $72.9_{\pm0.1}$ | $77.2_{\pm0.7}$ | $64.4_{\pm0.2}$ | $64.2_{\pm0.9}$ | $79.7_{\pm1.6}$ | $76.0_{\pm0.9}$ | $78.8_{\pm1.1}$ | $83.9_{\pm1.1}$ | 74.62 |
| | GEM | $72.4_{\pm0.4}$ | $78.1_{\pm0.1}$ | $69.2_{\pm0.4}$ | $67.2_{\pm0.4}$ | $90.1_{\pm1.3}$ | $81.7_{\pm0.5}$ | $80.6_{\pm0.9}$ | $85.6_{\pm1.1}$ | 78.11 |
| | UniMol | $72.9_{\pm0.6}$ | $79.6_{\pm0.5}$ | $69.6_{\pm0.1}$ | $65.9_{\pm1.3}$ | $91.9_{\pm1.8}$ | $82.1_{\pm1.3}$ | $80.8_{\pm0.3}$ | $85.7_{\pm0.2}$ | 78.56 |
| | C-FREE$_{Sch\text{-}1C}$ | $88.6_{\pm1.8}$ | $84.7_{\pm0.8}$ | $71.3_{\pm1.6}$ | $61.8_{\pm1.7}$ | $73.1_{\pm1.7}$ | $75.9_{\pm1.3}$ | $78.9_{\pm1.2}$ | $78.8_{\pm2.1}$ | 76.63 |
| | C-FREE$_{Sch\text{-}3C}$ | $78.9_{\pm1.1}$ | $84.2_{\pm0.4}$ | $71.7_{\pm0.9}$ | $62.5_{\pm1.9}$ | $83.7_{\pm2.9}$ | $82.5_{\pm0.1}$ | $77.9_{\pm1.2}$ | $78.6_{\pm1.1}$ | 77.50 |
| | C-FREE$_{PaiNN\text{-}3C}$ | $88.9_{\pm0.7}$ | $86.2_{\pm0.2}$ | $71.6_{\pm0.9}$ | $64.8_{\pm0.9}$ | $87.6_{\pm0.9}$ | $75.7_{\pm1.2}$ | $81.4_{\pm4.7}$ | $82.3_{\pm0.7}$ | 79.81 |

baselines, and (ii) property regression on the MARCEL benchmark, where we evaluate transferability to the Kraken dataset and study label efficiency on the larger Drugs-75K dataset.

### 4.2.1. Full Fine-tuning for Property Prediction

Building on the frozen backbone results, we next evaluate the adaptability of our models through end-to-end fine-tuning. We compare against the baselines reported in Yu et al. (2023), reproducing their evaluation protocol by attaching a linear classifier head and fine-tuning the entire backbone on each MoleculeNet dataset. We report results for a backbone using a PaiNN (Schütt et al., 2021) encoder, pre-trained with an ensemble of conformers (**C-FREE$_{PaiNN\text{-}3C}$**). For fairness, we also include a variant using SchNet (Schütt et al., 2017) (**C-FREE$_{Sch\text{-}3C}$**), as this is the 3D encoder used by the other baselines, as well as a variant pretrained with a single conformer (**C-FREE$_{Sch\text{-}1C}$**). As in Table 1, our backbones are pretrained on the GEOM dataset (Axelrod & Gomez-Bombarelli, 2022), which contains approximately 330K molecules. This contrasts with all other baselines, which are trained on the substantially larger PCQM4Mv2

*Table 3.* Performance on QM9 (Ramakrishnan et al., 2014) with full end-to-end fine-tuning, following the same protocol as in (Ji et al., 2024). In this setup, the full molecule is used as input to the downstream model. **C-FREE$_{\text{EGO}}$** and **C-FREE$_{\text{MURCKO}}$** denote the variant of C-FREE pretrained with EgoNets and Murcko segments respectively using only the 3D modality, while **C-FREE$_{\text{MM}}$** denotes the full multimodal variant used in Table 1. We additionally include **PaiNN (RND)**, a randomly initialized PaiNN model, as a baseline to disentangle the contribution of pretraining from PaiNN's inherent inductive bias. The evaluation metric is the Mean Absolute Error (MAE) (↓). Red highlights the best results and Blue the second best. **C-FREE** achieves the best results on 3 out of 6 datasets.

| METHOD | MU (↓) | ALPHA (↓) | HOMO/LUMO/GAP (↓) | $R^2$ (↓) | $C_v$ (↓) | ZPVE (↓) |
|---|---|---|---|---|---|---|
| GEM | $0.444_{\pm 1.5e-3}$ | $0.589_{\pm 4.2e-3}$ | $0.0067_{\pm 4e-5}$ | $25.67_{\pm 0.743}$ | $0.237_{\pm 1.4e-3}$ | $0.0011_{\pm 2.0e-5}$ |
| UNIMOL (1) | $0.155_{\pm 1.5e-3}$ | $0.363_{\pm 9.0e-3}$ | $0.0043_{\pm 2e-5}$ | $4.805_{\pm 0.055}$ | $0.183_{\pm 2.0e-3}$ | $0.0011_{\pm 3.0e-5}$ |
| UNIMOL2 310M | $0.092_{\pm 1.3e-3}$ | $0.315_{\pm 3.0e-3}$ | $0.0036_{\pm 1e-5}$ | $4.672_{\pm 0.245}$ | $0.143_{\pm 2.0e-3}$ | $0.0005_{\pm 1.0e-5}$ |
| UNIMOL2 1.1B | $0.089_{\pm 4.0e-4}$ | $0.305_{\pm 3.0e-3}$ | $0.0035_{\pm 1e-5}$ | $4.265_{\pm 0.067}$ | $0.144_{\pm 2.0e-3}$ | $0.0005_{\pm 1.0e-5}$ |
| **PAINN (RND)** | $0.042_{\pm 2.3e-3}$ | $0.149_{\pm 9.2e-3}$ | $0.0055_{\pm 2e-3}$ | $1.375_{\pm 0.223}$ | $0.061_{\pm 4.4e-3}$ | $0.0122_{\pm 2.5e-3}$ |
| **C-FREE$_{\text{3D-EGO}}$** | $0.064_{\pm 1.7e-4}$ | $0.116_{\pm 4.9e-4}$ | $0.0049_{\pm 7e-4}$ | $1.155_{\pm 0.010}$ | $0.049_{\pm 5.0e-4}$ | $0.0061_{\pm 4.1e-4}$ |
| **C-FREE$_{\text{3D-MURCKO}}$** | $0.077_{\pm 2.6e-3}$ | $0.124_{\pm 9.0e-3}$ | $0.0049_{\pm 5e-4}$ | $1.135_{\pm 0.025}$ | $0.052_{\pm 2.5e-3}$ | $0.0040_{\pm 2.0e-4}$ |
| **C-FREE$_{\text{MM}}$** | $0.057_{\pm 2.5e-3}$ | $0.107_{\pm 3.3e-3}$ | $0.0043_{\pm 1e-4}$ | $1.214_{\pm 0.068}$ | $0.044_{\pm 1.3e-3}$ | $0.0043_{\pm 9.0e-4}$ |

dataset from the OGB Large-Scale Challenge (Hu et al., 2021) (3M molecules), potentially placing our approach at a disadvantage.

We additionally include a suite of recent multimodal foundation models—MolFM (Luo et al., 2023b), GEM (Fang et al., 2022), and UniMol (Zhou et al., 2023)—and compare with their published results on the same downstream tasks. Note that these models are trained on substantially larger datasets: UniMol on 19M molecules aggregated from multiple sources, GEM on ZINC-20M, while MolFM relies on the PubChem dataset comprising 77M molecules. This setting tests how well the pretrained representations adjust to dataset-specific distributions and whether the multimodal backbone provides additional benefits. Performance is reported as mean ROC-AUC over three scaffold splits. A rundown and explanation of each baseline is found in the Appendix Section A.14.

As shown in Table 2, our method achieves notable gains on 5 of 8 datasets and delivers the best overall average when using the PaiNN variant. The strongest competitor, UniMol, also leverages multimodal inputs but was pretrained at a larger scale, namely on 19M molecules with roughly 209 million molecular conformations. Despite this large difference in pretraining scale, our approach matches or surpasses UniMol on several MoleculeNet tasks. We note that conformer generation fails for the larger molecules present in SIDER; we handle these cases by substituting dummy coordinates, which likely underlies C-FREE's underperformance on this dataset.

To further compare against larger-scale models, we conduct additional experiments on QM9 (Ramakrishnan et al., 2014) and ZINC (Gómez-Bombarelli et al., 2018). For QM9, we follow the setup in (Ji et al., 2024), restricting C-FREE to its 3D encoder and comparing it with the Uni-Mol (Ji et al., 2024) family, whose largest model contains 1.1 billion parameters. To isolate PaiNN's inductive bias, we add a randomly initialized PaiNN as baseline (PaiNN

(RND)); we also include results for the multimodal variant (C-FREE$_{\text{MM}}$) from Table 1 for completeness. Despite its substantially smaller size, our model outperforms Uni-Mol2 on four of the six prediction targets; the exceptions being HOMO/LUMO/GAP and ZPVE tasks, where Uni-Mol2 retains an advantage. C-FREE further outperforms PaiNN on five out of the six targets, with PaiNN only showing a slight edge on Mu, confirming that C-FREE's gains stem from pretraining rather than architectural bias alone. Among C-FREE variants, the multimodal version achieves the best overall performance, though the 3D-only variant holds a slight advantage on R2 and HOMO/LUMO/GAP. On ZINC, we compare our approach with GraphJEPA and find that it achieves superior performance. Further experimental details are provided in the Appendix Section A.13.

#### 4.2.2. COMPARISON WITH CHEMICALLY-INFORMED SUBGRAPH PRETRAINING

We perform a comparison of our pretrained backbone to a variant trained using Murcko-based fragments (Bemis & Murcko, 1996) on the QM9 (Ramakrishnan et al., 2014) dataset in Table 3. Murcko scaffolds capture the core ring systems of molecules and are commonly used to define chemically meaningful fragments. In this variant, the Murcko scaffold of each molecule serves as the target subgraph, while the remaining atoms and edges form the context, allowing us to assess the effect of using chemically informed fragments. We observe gains on the $R^2$ and ZPVE targets, whereas the EgoNet variant performs best on the remaining four tasks. Overall, these results suggest that our EgoNet-based approach already captures useful structural patterns without requiring hand-crafted chemical partitions, while remaining competitive across tasks.

#### 4.2.3. LABEL-EFFICIENT FINE-TUNING

Building on the observation that fine-tuning outperforms supervised training, we next study label efficiency on the Drugs-75K dataset (Zhu et al., 2024), a GEOM-Drugs subset with 75.099 molecules and at least 5 rotatable bonds. We

*Table 4.* Fine-tuning on the Drugs-75K dataset (Zhu et al., 2024) with limited labeled data. We compare our pretrained backbone (**FFT**) against a model trained from scratch (**RND**) using 1%, 10%, 50%, and 100% of the labels. pretraining offers clear gains in low-label regimes, while performance is comparable when using the full dataset.

| | | IP $\downarrow$ | EA $\downarrow$ | $\chi \downarrow$ |
|---|---|---|---|---|
| 1% | RND | $0.638_{\pm 0.001}$ | $0.613_{\pm 0.002}$ | $0.334_{\pm 0.001}$ |
| | FFT | $\mathbf{0.608}_{\pm 0.001}$ | $\mathbf{0.583}_{\pm 0.001}$ | $\mathbf{0.317}_{\pm 0.001}$ |
| 10% | RND | $0.561_{\pm 0.002}$ | $0.526_{\pm 0.001}$ | $0.277_{\pm 0.001}$ |
| | FFT | $\mathbf{0.520}_{\pm 0.005}$ | $\mathbf{0.494}_{\pm 0.002}$ | $\mathbf{0.267}_{\pm 0.001}$ |
| 50% | RND | $0.457_{\pm 0.002}$ | $0.433_{\pm 0.002}$ | $0.233_{\pm 0.001}$ |
| | FFT | $\mathbf{0.454}_{\pm 0.001}$ | $\mathbf{0.421}_{\pm 0.002}$ | $\mathbf{0.230}_{\pm 0.001}$ |
| 100% | RND | $\mathbf{0.419}_{\pm 0.005}$ | $0.403_{\pm 0.002}$ | $\mathbf{0.211}_{\pm 0.003}$ |
| | FFT | $\mathbf{0.419}_{\pm 0.002}$ | $\mathbf{0.395}_{\pm 0.003}$ | $0.213_{\pm 0.001}$ |

*Table 5.* Modality ablation on the Kraken dataset (MAE $\downarrow$). Using the same pretrained backbone, we compare fine-tuning (**FFT**) to random initialization (**RND**), feeding only the 2D sequence for the 2D variant and the 3D sequence for the 3D variant. pretraining consistently outperforms training from scratch. The multimodal model delivers the strongest performance, with 3D-only close behind, indicating that 3D features have greater impact than 2D.

| | | B5 $\downarrow$ | L $\downarrow$ | BurB5 $\downarrow$ | BurL $\downarrow$ |
|---|---|---|---|---|---|
| 2D | RND | $0.297_{\pm 0.006}$ | $0.396_{\pm 0.026}$ | $0.205_{\pm 0.006}$ | $0.152_{\pm 0.006}$ |
| | FFT | $0.276_{\pm 0.012}$ | $0.340_{\pm 0.028}$ | $0.176_{\pm 0.002}$ | $0.146_{\pm 0.005}$ |
| 3D | RND | $0.197_{\pm 0.006}$ | $0.345_{\pm 0.011}$ | $0.162_{\pm 0.006}$ | $0.135_{\pm 0.009}$ |
| | FFT | $\mathbf{0.194}_{\pm 0.003}$ | $0.329_{\pm 0.002}$ | $\mathbf{0.134}_{\pm 0.005}$ | $0.131_{\pm 0.004}$ |
| MM | RND | $0.203_{\pm 0.008}$ | $0.378_{\pm 0.003}$ | $0.161_{\pm 0.002}$ | $0.142_{\pm 0.001}$ |
| | FFT | $\mathbf{0.193}_{\pm 0.017}$ | $\mathbf{0.306}_{\pm 0.011}$ | $\mathbf{0.134}_{\pm 0.009}$ | $\mathbf{0.126}_{\pm 0.004}$ |

note that while these molecules appear in the pretraining set, the pretraining objective is self-supervised and label-free; the model is never exposed to any downstream labels during pretraining. The molecular overlap therefore does not constitute data leakage in any meaningful sense. For each molecule, Auto3D (Liu et al., 2022b) generates conformer ensembles, and three DFT-based reactivity descriptors serve as targets: ionization potential (IP), electron affinity (EA), and electronegativity ($\chi$).

We fine-tune our pretrained backbone using 1%, 10%, 50%, and 100% of the available data and compare against a model trained fully supervised from random initialization. With the full dataset, performance is broadly comparable, which is reasonable given the scale of the dataset. More importantly, in low-data regimes, initializing from self-supervised pretraining provides clear gains, consistently outperforming training from scratch (Table 4).

### 4.3. Modality Ablation

After establishing the benefits of our multimodal backbone, we next analyze the contribution of each modality through targeted ablations. Starting from the same GEOM-pretrained backbone, we fine-tune on Kraken while feeding either only the 2D encoder sequence, only the 3D sequence,

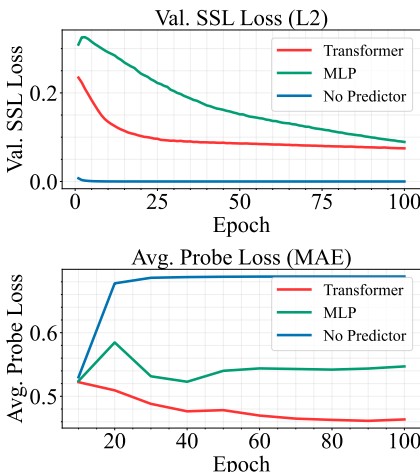

*Figure 2.* Predictor ablation. **Top**: SSL validation loss on GEOM during pretraining. **Bottom**: Average linear-probe MAE ($\downarrow$) on Kraken with frozen backbones. Without a predictor, training collapses (loss ~0) and probes perform worst; an MLP predictor helps but underperforms, while a Transformer predictor achieves the best results.

or both. This setup keeps the architecture and pretraining signal fixed, isolating the effect of each modality and mimicking transfer scenarios where only 2D or 3D data is available. Table 5 shows that pretraining consistently improves over random initialization across all settings, confirming that useful information is transferred even when restricted to a single modality. Among unimodal variants, the 3D-only backbone performs best, suggesting that geometric information has greater impact than 2D topology alone. Combining both modalities achieves the strongest results overall, reinforcing the view that 2D and 3D provide complementary signals.

### 4.4. Ablation on Predictor Types

We hypothesize that our model's strong performance stems from the predictor network, which serves as a guiding signal to refine the representations produced by the encoder. To test this, we pretrain on GEOM following the same setup previously described and perform an ablation with three predictor variants: none, a linear predictor, and a transformer. We then evaluate downstream performance using either a linear probe or full fine-tuning. To keep the study computationally lightweight, we use the smaller 2D-only pretrained backbone. Since the predictor design is identical across variants, this enables focused analysis without excessive computational cost.

As shown in Figure 2, removing the predictor leads to poor downstream performance across regression tasks, with the self-supervised loss collapsing to zero. While the EMA target encoder stabilizes training (Tarvainen & Valpola, 2017; Grill et al., 2020b; Assran et al., 2023), it is insufficient on its own: without an asymmetric architecture, the model

*Table 6.* EMA decay rate ablation with linear probing on the Kraken dataset (MAE $\downarrow$). We pretrain models using the same architecture, fixing the final decay rate $\tau_T = 1$ and varying the initial decay rates $\tau_0 \in \{1.0, 0.995, 0.9, 0.5\}$, where $\tau = [\tau_0, \tau_T]$ denotes the decay schedule. We include a randomly initialized model as baseline. Pretraining with $\tau_0 \neq 1.0$ consistently outperforms random initialization, and lower initial decay rates yield better results.

| | B5 $\downarrow$ | L $\downarrow$ | BURB5 $\downarrow$ | BURL $\downarrow$ | AVG |
|---|---|---|---|---|---|
| RND | $0.671_{\pm 0.001}$ | $0.667_{\pm 0.002}$ | $0.402_{\pm 0.002}$ | $0.242_{\pm 0.003}$ | 0.496 |
| $\tau_0 = 1.0$ | $0.684_{\pm 0.002}$ | $0.667_{\pm 0.005}$ | $0.411_{\pm 0.002}$ | $0.244_{\pm 0.001}$ | 0.502 |
| $\tau_0 = 0.995$ | $0.599_{\pm 0.009}$ | $0.593_{\pm 0.011}$ | $0.371_{\pm 0.003}$ | $0.245_{\pm 0.003}$ | 0.456 |
| $\tau_0 = 0.9$ | $0.583_{\pm 0.003}$ | $\mathbf{0.564_{\pm 0.002}}$ | $0.339_{\pm 0.003}$ | $0.243_{\pm 0.010}$ | 0.430 |
| $\tau_0 = 0.5$ | $\mathbf{0.571_{\pm 0.001}}$ | $0.575_{\pm 0.002}$ | $\mathbf{0.335_{\pm 0.001}}$ | $\mathbf{0.235_{\pm 0.003}}$ | **0.428** |

collapses to trivial solutions. Adding a predictor breaks this symmetry and prevents collapse (Richemond et al., 2020), consistent with theory showing that a trainable prediction head enables richer representations (Wen & Li, 2022). Even a simple MLP improves results, but the transformer performs best, likely because it operates at the node level before pooling, yielding more informative graph-level embeddings.

### 4.5. Ablation on EMA Decay Rate

Having established that the asymmetric nature of our architecture helps mitigate representational collapse during pretraining, we next turn to the EMA decay rate to study its influence on representation quality. For tractability, we pre-train on the smaller Drugs-75k dataset and extract a single 3-EgoNet subgraph per molecule. We train four models using $\tau_0 \in \{1.0, 0.995, 0.9, 0.5\}$, fixing $\tau_T = 1$ where $\tau = [\tau_0, \tau_T]$ denotes the decay schedule. Once converged, we checkpoint each model and evaluate via linear probing on the Kraken dataset following the same protocol as in Section 4.4.

As shown in Table 6, $\tau_0 = 1.0$ (no EMA) yields the worst performance, suggesting the model fails to learn meaningful representations without a momentum teacher, likely due to training loss collapse (see Section A.6). Lower decay values consistently improve performance, suggesting that loosening the coupling between teacher and context encoder encourages more diverse target representations. While lower decay rates achieve superior results, we adopt the more conservative $\tau_0 = 0.995$ in our main experiments to prioritize training stability; see Section A.6 for details.

### 4.6. Ablation on k-EgoNet

To provide insight into the effect of the ego-net radius parameter $k$, we compute structural features on subgraphs generated for $k = \{1, 2, 3, 4, 5\}$ over 200 randomly sampled molecules from the GEOM dataset, summarized in Appendix A.10. Alongside this analysis, we ablate $k$ directly by pretraining models with 1-, 3-, and 5-EgoNets, keeping the architecture and training setup fixed as in Section 4.5.

*Table 7.* Ablation on the k-EgoNet radius parameter $k$, with linear probing on the Kraken dataset (MAE $\downarrow$). Using the same architecture, we pretrain models extracting only a single k-EgoNet per molecule for $k \in \{1, 3, 5\}$ and include a randomly initialized model as baseline. The best performance is achieved by $k = 5$, suggesting that the model learns the best representations for comparable sized context and target subgraphs, $k = 1$ shows a similar performance to the randomly initialized model, suggesting that overly local subgraphs do not capture sufficient structural signal to guide meaningful representation learning.

| | B5 $\downarrow$ | L $\downarrow$ | BURB5 $\downarrow$ | BURL $\downarrow$ | AVG |
|---|---|---|---|---|---|
| RND | $0.671_{\pm 0.001}$ | $0.667_{\pm 0.002}$ | $0.402_{\pm 0.002}$ | $0.242_{\pm 0.003}$ | 0.496 |
| $k = 1$ | $0.677_{\pm 0.001}$ | $0.648_{\pm 0.009}$ | $0.444_{\pm 0.001}$ | $0.239_{\pm 0.002}$ | 0.491 |
| $k = 3$ | $0.599_{\pm 0.009}$ | $0.593_{\pm 0.011}$ | $0.371_{\pm 0.006}$ | $0.245_{\pm 0.003}$ | 0.456 |
| $k = 5$ | $0.558_{\pm 0.002}$ | $0.601_{\pm 0.014}$ | $0.365_{\pm 0.001}$ | $0.242_{\pm 0.002}$ | 0.438 |

Table 7 demonstrates that pretraining with $k = 5$ achieves the best performance, suggesting that the model learns richer representations when the target and context views are of comparable size. Conversely, $k = 1$ yields the worst performance, indicating that overly local subgraphs do not capture sufficient structural signal to guide meaningful representation learning.

## 5. Conclusions and Future Work

We introduced C-FREE, a contrast-free multimodal self-supervised framework for molecular representation learning. Its core idea is to align embeddings of complementary subgraphs, enabling predictive pretraining without negatives, positional encodings, or costly clustering. By combining 2D molecular graphs and 3D conformer ensembles, C-FREE achieves state-of-the-art results on classification (MoleculeNet) and regression (Kraken, Drugs-75K and QM9), with clear gains in low-label regimes. Ablations show that the EMA scheduler and the predictor are critical to avoid collapse and that 2D and 3D provide complementary signals, with 3D offering stronger performance but 2D remaining competitive. Finally, experiments on the EXP benchmark confirm that C-FREE is more expressive than 1-WL.

Looking ahead, several extensions are promising. Our largest model has only 9.1M parameters and was pretrained on 0.33M molecules; scaling to larger architectures and datasets, as the 100M-molecule collections in (Beaini et al., 2024), could unlock further gains. Another extension is combining self-supervised pretraining with an additional supervised stage on such large-scale data before transferring to downstream tasks. Extending to new modalities, such as SMILES strings, is also appealing, though it may require modality-specific masking. Furthermore, our encoder-agnostic framework naturally supports encoders with domain-specific inductive biases, such as chirality-aware models, which can either replace or complement existing encoders as an additional modality. Finally, more chemically informed subgraph selection or alternative alignment objectives could further enrich representations.

## Acknowledgements

The authors would like to thank the anonymous reviewers for their insightful feedback. The authors acknowledge funding by the Ministry of Science, Research and the Arts Baden-Württemberg via the Artificial Intelligence Software Academy (AISA). AM and MN acknowledge the support from the International Max Planck Research School for Intelligent Systems (IMPRS-IS). AM acknowledges funding by the EU Horizon project ELIAS (No. 101120237). BA acknowledges support from the ELLIS Unit Stuttgart. The authors are also grateful to Samir Darouich for his assistance with code development and for his chemistry expertise, which helped address technical questions that arose during this work.

## Impact Statement

This paper presents work aimed at advancing molecular representation learning, with direct applications to drug discovery and computational chemistry. By improving the quality of learned molecular representations, our method has the potential to accelerate the early stages of drug development, including hit identification and lead optimization. We do not foresee any immediate negative societal consequences of this work.

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

# A. Supplementary Materials

## A.1. Expressiveness and Invariance

We preserve the invariance properties of the encoders used in our framework. In particular, we restate the result of Manolache et al. (2024) for our setting, restricted to 2D graphs and 3D conformers:

**Lemma A.1.** *Let $G$ be a 2D molecular graph and $\{c_1, \ldots, c_k\}$ a set of $k$ 3D conformers for the same molecule. Let $\hat{y} = f_\theta(G, \{c_1, \ldots, c_k\})$ be the encoder output as defined in Section 3. Suppose the 3D encoder is invariant to the actions of a group $\mathcal{G}$. Then $f_\theta$ is also invariant to any $T_1, \ldots, T_k \in \mathcal{G}$, i.e.,*

$$f_\theta(G, \{T_1 c_1, \ldots, T_k c_k\}) = f_\theta(G, \{c_1, \ldots, c_k\}).$$

The proof of this result is given in Manolache et al. (2024); it applies directly in our case after removing the SMILES modality.

Next, we provide the proof for Lemma 1 in the main paper.

**Lemma A.2.** *Let C-FREE$_{DS}$ be a model as defined in Section 3 with a subgraph encoder $f_\theta$ consisting of a 1-WL MPNN (e.g., GIN/GINE) followed by a Transformer without positional encodings, and a DeepSets task head DS. For any $k$-EgoNet policy with $k \geq 1$ under the assumptions of Theorem 2 from (Bevilacqua et al., 2022), C-FREE$_{DS}$ is as expressive as ESAN (Bevilacqua et al., 2022) with an EGO policy, therefore it is at most as expressive as DS-WL and strictly more expressive than 1-WL.*

*Proof.* Fix a $k$-EGO policy $\pi = \text{EGO}_k$ with $k \geq 1$ and let $S_\pi(G)$ be the multiset of $k$-ego-nets with their complements (edge-covering in the ESAN sense). Define:

$$f_{C\text{-}FREE}(G) = DS\big(\{f_\theta(S) : S \in S_\pi(G)\}\big),$$

Due to Theorem A.1, we have that $f_\theta$ maintains the permutation equivariance of the MPNN; moreover, since there exists a parametrization of the Transformer that can approximate the identity map arbitrarily well, the Transformer does not lower the expressive power of the MPNN. We therefore have that $f_\theta$ is as powerful as 1-WL.

Since we have a DeepSets encoder $DS$ and an edge-covering $k$-EGO policy, we can use the same proof argument as in Theorem 2 from (Bevilacqua et al., 2022), i.e. we apply $f_\theta$ to each $S \in S_\pi(G)$ and then aggregate the multisets with $DS$, therefore $f_{C\text{-}FREE}$ simulates ESAN, and is at most as expressive as *DS-WL* and strictly more expressive than 1-WL.

$\square$

## A.2. Theory Alignment

To validate our theoretical findings with an experiment on the EXP dataset, designed by Abboud et al. (2021) such that any 1-WL GNN cannot exceed random guessing. We train a smaller version of our encoder and compare it against GINE (Hu et al., 2020a) and GraphJEPA (Assran et al., 2023). Results are averaged over three runs with resampled EgoNets. As shown in Table 8, even the 1-EgoNet variant of C-FREE approaches the theoretical upper bound, while 2- and 3-EgoNet variants achieve the highest accuracy, outperforming both GINE and GraphJEPA, the latter relying on the costly METIS (Karypis & Kumar, 1998) algorithm. Nevertheless, the practical significance of expressiveness for molecular learning is limited, as recent work (Pellizzoni et al., 2025) shows that 1-WL already distinguishes most samples in molecular datasets nearly perfectly. Although less decisive for molecules, expressiveness may be more important in other domains, where the flexibility of our framework could be advantageous.

## A.3. Target and Context Views Generation using k-EgoNets

An example of context and target views is shown in Figure 3, where the Oxygen molecule serves as the anchor and its 2-EgoNet is extracted.

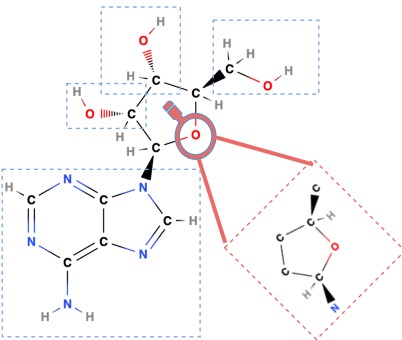

*Figure 3.* To generate subgraphs, we sample a random node from the original graph (here, the oxygen atom) and extract its 2-EgoNet as the context subgraph (outlined by red square). The remaining components (outlined by blue squares) constitute the target subgraph.

*Table 8.* Expressiveness results on EXP (Abboud et al., 2021), where 1-WL GNNs cannot surpass random guessing. Even the 1-EgoNet variant of C-FREE approaches the theoretical upper bound, while 2- and 3-EgoNet variants achieve the highest accuracy, outperforming GINE (Hu et al., 2020a) and GraphJEPA (Assran et al., 2023). Unlike GraphJEPA, our method avoids costly METIS clustering (Karypis & Kumar, 1998).

| Method | Accuracy ($\uparrow$) |
|---|---|
| GINE | $50.69_{\pm 1.39}$ |
| GraphJEPA | $98.77_{\pm 0.99}$ |
| C-FREE (1-Ego) | $96.03_{\pm 1.22}$ |
| C-FREE (2-Ego) | $\mathbf{99.33}_{\pm 0.18}$ |
| C-FREE (3-Ego) | $99.08_{\pm 0.20}$ |

*Table 9.* Performance on ZINC (Gómez-Bombarelli et al., 2018) with full end-to-end fine-tuning. C-FREE outperforms GraphJEPA despite not relying on positional encoding.

| MODEL | ZINC ($\downarrow$) |
|---|---|
| GRAPHJEPA | $0.434_{\pm 0.014}$ |
| **C-FREE (OURS)** | $\mathbf{0.204}_{\pm 0.015}$ |

### A.4. Exclusion of 1D Modality

While SMILES strings are included as a third modality in Manolache et al. (2024), we chose to exclude them here for both practical and conceptual reasons. Practically, our self-supervised objective masks subgraphs defined by graph topology via $k$-EgoNets, and since SMILES is a sequential serialization of the molecular graph, there is no clean correspondence between a masked subgraph and a contiguous region of the SMILES string. Designing a masking strategy that consistently occludes the same chemical information across all modalities, without leaking structural cues into the remaining views, is therefore non-trivial. Conceptually, SMILES and 2D molecular graphs encode largely redundant information, a finding consistent with the supervised multimodal setting of Manolache et al. (2024), where the 1D modality contributed least to overall performance.

### A.5. Ablation on Separator Tokens

Given the permutation equivariance of the transformer, separator tokens are not strictly necessary. Our design choice was primarily motivated by Manolache et al. (2024), whose formulation we followed for consistency, as the computational cost is negligible. However, we hypothesize that separator tokens may serve as attention sinks (Xiao et al., 2024), which could be beneficial in our setting since we do not use positional encodings. To investigate this, we conducted an ablation study using the setup described in 4.5, training two model variants: one with separator tokens and one without. Results in Table 10 show a slight but statistically insignificant improvement with separator tokens.

### A.6. Ablation on EMA decay rate

We complement the decay rate ablation with Figure 4, which presents the pretraining loss across different initial decay rates. The training loss exhibits notable sensitivity to the choice of the EMA decay rate. All three variants begin with identical loss values but diverge during training. A high decay rate ($\tau_0 = 1$) causes abrupt loss collapse after the first epoch, after which the loss remains static. In contrast, a lower decay rate ($\tau_0 = 0.8$) produces noisier but faster convergence. The standard C-FREE setting ($\tau_0 = 0.995$) achieves slower yet smoother convergence.

### A.7. Additional Details on Empirical Evaluation

For the multimodal variant, the context encoder consists of three components: a 6-layer GINE (Xu et al., 2019) with hidden dimension 128, a PaiNN (or SchNet) with hidden dimension 128, 6 interaction steps, and a cutoff of 10, and 6 Transformer layers with 8 heads and hidden dimension 512. For the pretrained 2D variant, we use the same GINE configuration and the same number of Transformer layers and heads, but reduce the hidden dimension to 387. In both variants, the predictor is implemented as 4 Transformer layers with 4 heads each. The parameters are updated via backpropagation using the Adam optimizer, while the target encoder is updated through an exponential moving average (EMA) schedule, with the decay rate $\tau_t$ gradually increasing from 0.995 to 1.0 over the course of training.

*Table 10.* Separator tokens ablation with linear probing on the Kraken dataset (MAE ↓). Models were pretrained using identical architectures, with one variant including separator tokens between modalities in the transformer sequence and another without. Results show a modest improvement with separator tokens, suggesting they may function as attention sinks.

|  | KRAKEN (AVG) |
|---|---|
| WITH | $0.405_{\pm 0.010}$ |
| WITHOUT | $\mathbf{0.449_{\pm 0.005}}$ |

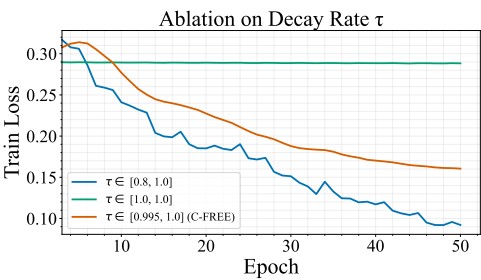

*Figure 4.* EMA decay rate ablation: SSL pretraining loss on GEOM. Models with no EMA exhibit training collapse, while slower EMA decay rates achieve better convergence with increased noise.

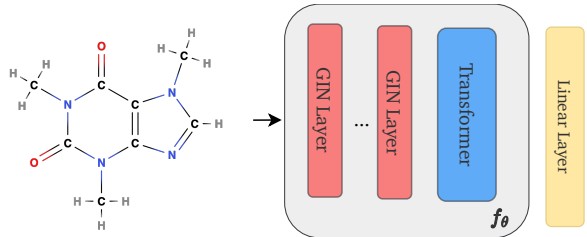
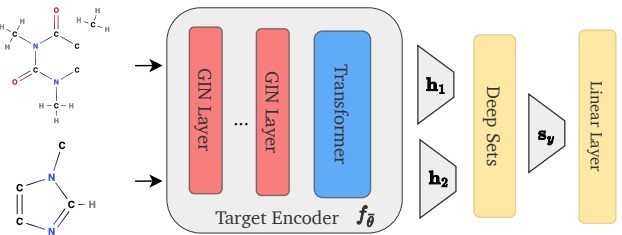

*Figure 5.* Fine-tuning strategies for C-FREE. Left: C-FREE$_{\text{MOL}}$: use the pretrained encoder to produce a whole-molecule embedding and attach a lightweight task head (e.g., linear probe or small MLP) for downstream prediction. Right: C-FREE$_{\text{DS}}$: aggregate multiple $k$-EgoNet subgraph embeddings with a DeepSets aggregator and apply a lightweight task head to the aggregated representation. For evaluation we typically use a linear probe on a frozen backbone to assess representation quality; for downstream tasks (regression, classification) the probe can be replaced by a task-specific head and the backbone optionally fine-tuned.

Since the EMA decay reaches $\tau_t = 1$ in the final epoch, the context and target encoders converge to identical parameters. Nevertheless, we follow (Assran et al., 2023) and report results using the target encoder.

For the choice of the scheduler we opt for a cosine scheduler without warmup. We notice that using a very small learning rate prevented convergence, while a moderate learning rate caused an early loss drop followed by stagnating representations. Adding a warmup phase allows the model to adapt gradually before the cosine decay, improving stability and representation learning. Thus we begin with a learning rate of $2 \times 10^{-6}$ over 30 epochs warmup up to $5 \times 10^{-5}$ and a patience of 50 epochs. For the batch size we use 256 and a weight decay of 0.04 and train for 200 epochs.

All experiments were performed on a mix of Nvidia A100/RTX 4090 GPUs and AMD EPYC 7713/Intel Xeon W-2225 CPUs for both the pretraining and downstream experiments. All experiments consumed a total of approximately 500 GPU hours, with the longest compute being consumed on the pretraining backbone run on the GEOM dataset with a total of 25 hours.

### A.8. Downstream Pipeline

In Figure 5, we illustrate the downstream pipeline. We retain the target encoder from pretraining and attach a prediction head. For the linear probe using the full molecule, we feed the entire molecule into the encoder, perform a single forward pass, and apply a linear head for prediction. For the subgraph-based variant, each subgraph is processed independently by the encoder; their representations are then aggregated using a DeepSets module, and the resulting aggregated embedding is passed through a linear head for the final prediction.

### A.9. Backbone Parameter Efficiency

Table 11 compares the number of trainable parameters across different SSL backbones. For our method, we report both the total parameters and the encoder-only count used during downstream evaluation. This distinction arises because only the target encoder is retained as the backbone, during downstream tasks, while the predictor is discarded. As a result, nearly half of the parameters are removed at this stage, allowing our method to remain competitive without increasing the parameter load for downstream evaluation, further underscoring its efficiency.

*Table 11.* Computational efficiency of different SSL methods from (Wang et al., 2023a), showing the number of trainable parameters for each backbone. We report both the total parameters of our backbone and those of the encoder alone, since only the latter is used for downstream evaluation. By discarding nearly half of the backbone parameters in this stage, our approach remains competitive without increasing the parameter count for downstream tasks, further highlighting its efficiency.

| METHOD | #PARAMETERS (MILLION) |
|---|---|
| EDGEPRED | 7.46 |
| ATTRMASK | 7.61 |
| GPT-GNN | 7.61 |
| INFOGRAPH | 7.82 |
| GROVER | 7.57 |
| CONT.PRED | 12.00 |
| GRAPHCL | 8.19 |
| JOAO | 8.19 |
| GRAPHMVP | 15.84 |
| **C-FREE$_{2D}$ (FULL)** | 8.09 |
| **C-FREE$_{2D}$ (ENCODER)** | 4.67 |
| **C-FREE$_{MM}$ (FULL)** | 14.65 |
| **C-FREE$_{MM}$ (ENCODER)** | 9.12 |

*Table 12.* Mean structural features of context subgraphs for different ego-net sizes $k$.

| | $\mu_{atoms}$ | $\mu_{bonds}$ | $\mu_{deg}$ | $\mu_{cov}$ |
|---|---|---|---|---|
| $k = 1$ | 2.867 | 1.867 | 0.603 | 0.071 |
| $k = 2$ | 7.133 | 6.133 | 0.849 | 0.176 |
| $k = 3$ | 10.600 | 10.000 | 0.933 | 0.258 |
| $k = 4$ | 15.600 | 15.533 | 0.989 | 0.385 |
| $k = 5$ | 21.067 | 21.200 | 1.003 | 0.524 |

*Table 13.* Overview of tasks and sizes for the MoleculeNet datasets.

| | BBBP | TOX21 | TOXCAST | SIDER | CLINTOX | MUV | HIV | BACE |
|---|---|---|---|---|---|---|---|---|
| # MOLECULES | 2,039 | 7,831 | 8,575 | 1,427 | 1,478 | 93,087 | 41,127 | 1,513 |
| # TASKS | 1 | 12 | 617 | 27 | 2 | 17 | 1 | 1 |

*Table 14.* Overview of tasks and sizes for the GEOM, Drugs-75K and Kraken datasets.

| | GEOM | DRUGS-75K | KRAKEN |
|---|---|---|---|
| # MOLECULES | 304,466 | 75,099 | 1,552 |
| # CONFORMERS | 25M | 558,002 | 21,287 |
| # TASKS | - | 3 | 4 |

## A.10. Analysis of Subgraphs for Different Ego-Net Sizes

To provide insight into the effect of ego-net radius $k$, we compute structural features on subgraphs generated for $k = \{1, 2, 3, 4, 5\}$ over 200 randomly sampled molecules from the GEOM dataset. The sampled molecules have an average size of $\mu_{atoms} = 42$ and an average connectivity of $\mu_{deg} = 1.043$. Table 12 reports, for each value of $k$, the mean structural features aggregated over all context subgraphs: the number of atoms ($\mu_{atoms}$), the number of bonds ($\mu_{bonds}$), the connectivity defined as the ratio of bonds to atoms ($\mu_{deg}$), and the coverage defined as the fraction of atoms in the context subgraph relative to the full molecule ($\mu_{cov}$).

We observe that for $k = 3, 4$, the context subgraphs achieve a coverage of 26-52%, most likely capturing meaningful local chemical environments, such as a functional group together with their immediate neighbors, while leaving enough information in the complementary subgraph to predict the missing signal.

## A.11. Tasks and Dataset Sizes Overview

In Table 13, we summarize the tasks and dataset sizes of the MoleculeNet benchmarks. Likewise, in Table 14, we present an overview of the GEOM, Drugs-75K, and Kraken datasets, including the corresponding number of conformers.

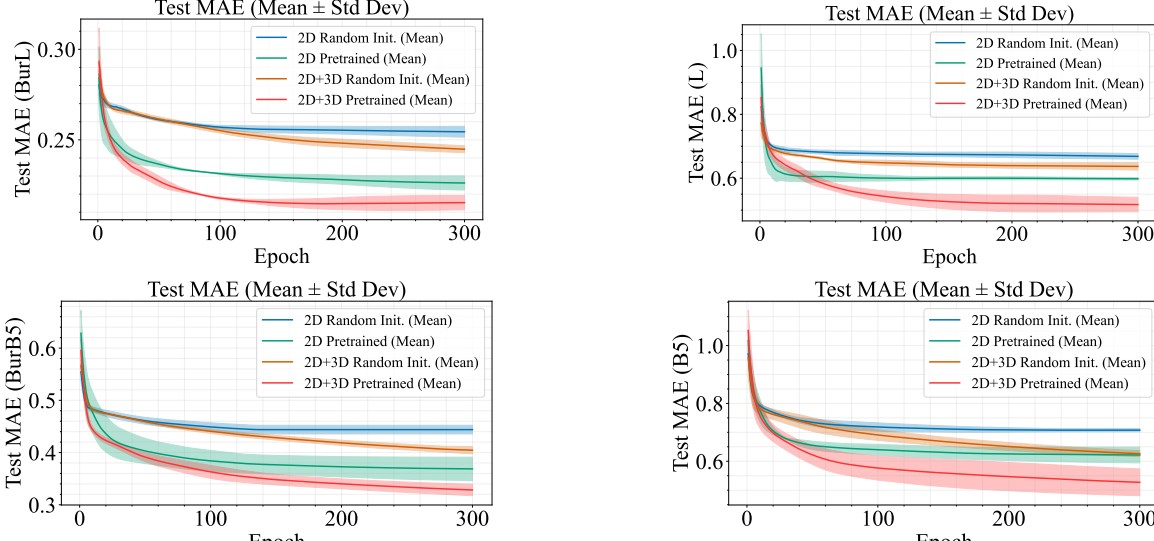

*Figure 6.* Test MAE on the Kraken regression tasks (Sterimol BurL and Sterimol L) with frozen backbones. GEOM-pretrained models consistently outperform random initialization for both 2D-only and multimodal variants. pretrained models begin with lower error and converge faster, while randomly initialized models fail to match performance even after 300 epochs. Incorporating the 3D modality yields further gains over 2D-only backbones, with pretraining amplifying this advantage. Curves show the mean over 3 runs, with shaded regions indicating the standard deviation.

*Table 15.* Performance on MoleculeNet ([Wu et al., 2018](#)) with 2D-only frozen backbones. **Non-CL** denotes non-contrastive and **CL** contrastive methods. We report **C-FREE$_{2D}$** (2D-only) and **C-FREE$_{MM}$** (multimodal), each with linear probing on whole-molecule embeddings (**MOL**) or on subgraphs using subgraph aggregation with DeepSets ([Zaheer et al., 2017](#)) (**SUB**). Metric: ROC-AUC (↑). Red marks the best model and Blue the second best. **C-FREE** ranks first or second on 6 of 8 datasets, with **MM-MOL** best overall, while even the 2D-only variants of C-FREE outperform all baselines on average.

| | | **MoleculeNet Datasets (Linear Probe)** | | | | | | | |
| | | BBBP (↑) | Tox21 (↑) | ToxCast (↑) | Sider (↑) | ClinTox (↑) | MUV (↑) | HIV (↑) | Bace (↑) | Avg (↑) |
|---|---|---|---|---|---|---|---|---|---|---|
| | Random init. | $50.7_{\pm2.5}$ | $64.9_{\pm0.5}$ | $53.2_{\pm0.3}$ | $53.2_{\pm1.1}$ | $63.1_{\pm2.3}$ | $62.1_{\pm1.3}$ | $66.1_{\pm0.7}$ | $63.4_{\pm1.8}$ | 59.60 |
| **CL** | InfoGraph | $65.9_{\pm0.6}$ | $65.8_{\pm0.7}$ | $54.6_{\pm0.1}$ | $57.2_{\pm1.0}$ | $61.4_{\pm4.8}$ | $63.9_{\pm1.9}$ | $71.4_{\pm0.6}$ | $67.4_{\pm4.9}$ | 63.44 |
| | Grover | $67.0_{\pm0.3}$ | $63.9_{\pm0.3}$ | $53.6_{\pm0.4}$ | $59.9_{\pm1.7}$ | $65.0_{\pm6.4}$ | $62.7_{\pm1.4}$ | $67.8_{\pm1.0}$ | $69.0_{\pm4.7}$ | 63.62 |
| | GraphCL | $64.7_{\pm1.7}$ | $69.1_{\pm0.5}$ | $56.2_{\pm0.2}$ | $59.5_{\pm0.9}$ | $60.8_{\pm3.0}$ | $60.6_{\pm1.8}$ | $72.5_{\pm1.4}$ | $77.0_{\pm1.7}$ | 65.04 |
| | JOAO | $66.1_{\pm0.8}$ | $68.1_{\pm0.2}$ | $55.1_{\pm0.4}$ | $58.3_{\pm0.3}$ | $65.3_{\pm6.1}$ | $62.4_{\pm1.2}$ | $73.8_{\pm1.2}$ | $71.1_{\pm0.8}$ | 65.05 |
| | GraphMVP | $69.2_{\pm1.8}$ | $63.8_{\pm0.3}$ | $55.5_{\pm0.3}$ | $58.6_{\pm0.4}$ | $58.7_{\pm1.9}$ | $63.8_{\pm1.3}$ | $68.6_{\pm1.0}$ | $73.3_{\pm4.7}$ | 63.92 |
| **Non-CL** | EdgePred | $54.2_{\pm1.0}$ | $66.2_{\pm0.2}$ | $54.4_{\pm0.1}$ | $56.1_{\pm0.1}$ | $65.4_{\pm5.0}$ | $59.5_{\pm0.9}$ | $73.6_{\pm0.4}$ | $71.4_{\pm1.2}$ | 62.59 |
| | AttrMask | $62.7_{\pm2.7}$ | $65.7_{\pm0.8}$ | $56.1_{\pm0.2}$ | $58.3_{\pm1.5}$ | $61.9_{\pm6.4}$ | $60.9_{\pm1.8}$ | $65.5_{\pm1.4}$ | $64.8_{\pm2.6}$ | 61.99 |
| | GPT-GNN | $62.0_{\pm0.9}$ | $64.9_{\pm0.7}$ | $55.4_{\pm0.2}$ | $55.3_{\pm0.8}$ | $55.0_{\pm5.1}$ | $61.2_{\pm1.5}$ | $71.2_{\pm1.5}$ | $61.0_{\pm1.2}$ | 60.74 |
| | ContextPred | $55.5_{\pm2.0}$ | $67.9_{\pm0.7}$ | $54.0_{\pm0.3}$ | $57.1_{\pm0.5}$ | $67.4_{\pm4.3}$ | $60.5_{\pm0.9}$ | $66.2_{\pm1.5}$ | $54.4_{\pm3.2}$ | 60.36 |
| | **C-FREE$_{2D-MOL}$** | $60.5_{\pm1.7}$ | $76.1_{\pm0.2}$ | $62.7_{\pm0.4}$ | $59.0_{\pm0.6}$ | $62.7_{\pm1.0}$ | $67.6_{\pm0.5}$ | $68.7_{\pm0.4}$ | $75.8_{\pm0.9}$ | 66.63 |
| | **C-FREE$_{2D-SUB}$** | $64.2_{\pm3.8}$ | $76.7_{\pm0.6}$ | $63.9_{\pm0.3}$ | $58.0_{\pm0.7}$ | $71.4_{\pm3.7}$ | $64.6_{\pm3.1}$ | $65.5_{\pm0.6}$ | $73.9_{\pm0.7}$ | 67.27 |
| | **C-FREE$_{MM-MOL}$** | $69.8_{\pm2.6}$ | $79.9_{\pm1.1}$ | $65.8_{\pm0.7}$ | $58.5_{\pm2.5}$ | $69.9_{\pm1.9}$ | $76.6_{\pm2.8}$ | $72.8_{\pm0.7}$ | $75.3_{\pm1.1}$ | 71.07 |
| | **C-FREE$_{MM-SUB}$** | $73.8_{\pm2.1}$ | $76.7_{\pm0.7}$ | $66.8_{\pm0.2}$ | $56.4_{\pm1.5}$ | $75.7_{\pm2.2}$ | $70.6_{\pm1.0}$ | $71.9_{\pm1.5}$ | $75.5_{\pm1.9}$ | 70.92 |

## A.12. Comparison with Frozen Backbones on the Kraken Dataset

We evaluate transfer to molecular property regression on Kraken ([Gensch et al., 2022](#)), which contains 1,552 ligands labeled with four 3D descriptors (Sterimol B5, Sterimol L, buried Sterimol B5, buried Sterimol L). Since Kraken molecules are disjoint from GEOM, it provides a strong test of generalization. As shown in Figure 6, GEOM-pretrained backbones start with lower error, converge faster, and consistently outperform random initialization, with even the 2D-only pretrained model surpassing a randomly initialized multimodal one. Adding 3D information yields further gains, amplified by pretraining.

## A.13. Additional Details and Results

On the ZINC benchmark (Table 9), our method shows a clear performance improvement over GraphJEPA. This suggests that the gains come from the combination of architectural choices and the overall training pipeline, with EgoNets contributing as one part of this broader design.

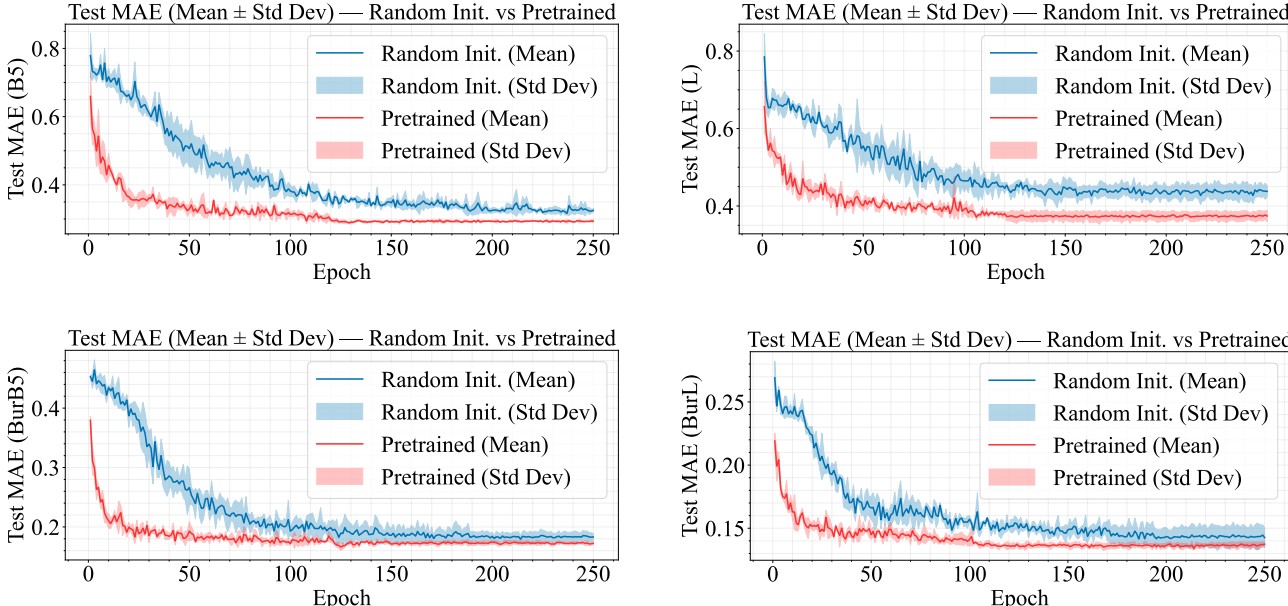

*Figure 7.* Test MAE on Kraken regression tasks (Sterimol L,B5, BurB5 and BurL) comparing random initialization and 2D-only GEOM-pretrained models. pretrained models start with lower error and converge faster, while randomly initialized models fail to match their performance even after 250 epochs. Curves show the mean over 3 runs, with shaded regions indicating the standard deviation.

*Table 16.* Ablation study on the Kraken dataset (MAE ↓). We keep the encoder fixed and compare three predictors: (1) none, (2) a 2-layer MLP, and (3) a transformer. The transformer consistently achieves the best performance. The gap is especially pronounced in the linear probe (LIN. P.) setting, where the quality of the learned representations matters most. Even with full fine-tuning (FT), the no-predictor and MLP variants fail to match the transformer predictor.

|  | METHOD | B5 ↓ | L ↓ | BURB5 ↓ | BURL ↓ |
|---|---|---|---|---|---|
| FT | NONE | $0.381_{\pm 0.023}$ | $0.494_{\pm 0.020}$ | $0.202_{\pm 0.009}$ | $0.157_{\pm 0.004}$ |
|  | 2-LAYERS MLP | $0.315_{\pm 0.017}$ | $0.396_{\pm 0.018}$ | $0.185_{\pm 0.009}$ | $\mathbf{0.144}_{\pm 0.004}$ |
|  | TRANSFORMER | $\mathbf{0.292}_{\pm 0.006}$ | $\mathbf{0.380}_{\pm 0.023}$ | $\mathbf{0.180}_{\pm 0.014}$ | $0.146_{\pm 0.004}$ |
| LIN. P. | NONE | $1.065_{\pm 0.001}$ | $0.814_{\pm 0.001}$ | $0.624_{\pm 0.001}$ | $0.296_{\pm 0.001}$ |
|  | 2-LAYERS MLP | $0.817_{\pm 0.002}$ | $0.687_{\pm 0.008}$ | $0.514_{\pm 0.001}$ | $0.266_{\pm 0.001}$ |
|  | TRANSFORMER | $\mathbf{0.588}_{\pm 0.004}$ | $\mathbf{0.554}_{\pm 0.007}$ | $\mathbf{0.347}_{\pm 0.003}$ | $\mathbf{0.202}_{\pm 0.008}$ |

In Table 15 we provide the full linear probe results paper from (Wang et al., 2023a). In addition, we also report results for full fine-tuning on the Kraken dataset. For this, we use the same setup as in Section 4.1, with an MLP head, and fine-tune the model end-to-end. In Figure 7, we show results for the 2D-only variant to highlight the initial performance gap, where the pretrained backbone converges faster than the randomly initialized one.

Finally, in Table 16, we report the explicit numerical results of the probe presented in Section 4.4, evaluated after the full convergence of the model. Additionally, we include the results from end-to-end fine-tuning using the same experimental setup, providing a complete view of how the model performs when the entire backbone is updated. These fine-tuning results further confirm our observation that the predictor transformer consistently outperforms the MLP predictor, while omitting the predictor altogether leads to substantially worse performance, highlighting the importance of the predictor design in our framework.

## A.14. Additional Related Work

**Contrast-Free** In this paper, our use of the term "contrast-free" follows the standard terminology in the representation learning literature, where methods such as BYOL (Grill et al., 2020b), DINO (Oquab et al., 2023), and JEPA (Assran et al., 2023) are described as contrast-free because they avoid explicit contrastive objectives, large negative sets, and the large batch requirements of InfoNCE-style losses. We use the term in the same sense here: our model does not rely on negative pairs, large batch sizes, or explicit contrastive objectives, which helps avoid the computational overhead and instability often

seen in contrastive training.

**Contrastive Learning** UniCorn (Feng et al., 2024) presents a unified contrastive learning framework that integrates multiple molecular views and existing methods into a single pretraining approach. 3D-Mol (Kuang et al., 2024) leverages 3D conformational information by constructing hierarchical graphs and applying contrastive learning to differentiate molecular conformations. GraphFP (Luong & Singh, 2023) captures higher-level connectivity through fragments (representations of molecular substructures) aligning fragment embeddings with their corresponding graph regions to enable multi-resolution structural learning. MolCLR (Wang et al., 2022) employs three graph augmentations, namely atom masking, bond deletion, and subgraph removal and uses contrastive learning to bring augmented views of the same molecule closer, and Galformer (Bai et al., 2023) applies dual-view contrastive learning. Finally, GraphLoG (Xu et al., 2021) captures both local similarities and global semantic clusters in whole-graph representations using hierarchical prototypes trained via an online EM algorithm.

**Generative Learning** GraphMAE (Hou et al., 2022) adapts the Masked Autoencoder (MAE) from the vision domain to graphs, focusing on reconstructing node attribute features using a scaled cosine error. Similarly, MoleBERT (Xia et al., 2023) extends this idea with a VQ-VAE-based context-aware tokenizer that encodes atom attributes into a larger, chemically meaningful discrete vocabulary, enabling a Masked Atoms Modeling (MAM) task where GNNs predict masked atom codes rather than raw features. Finally, MGSSL (ZHANG et al., 2021) leverages a BRICS-based fragmentation method to extract molecular motifs and pretrains GNNs to predict motif topology and labels, incorporating multi-level pretraining to capture both local and global graph information.

**Latent Representation Learning** CCA-SSG (Zhang et al., 2021) introduces an alignment objective based on Canonical Correlation Analysis, encouraging the latent features of two augmented views to be maximally correlated while remaining de-correlated across dimensions. Complementing these augmentation-based methods, AFGRL (Lee et al., 2022) proposes a more principled strategy for view generation by identifying structurally and semantically similar anchor nodes, mitigating the reliance on handcrafted augmentations.

**Molecular Foundation Models** Early SMILES-only models such as ChemBERTa-v1 (Chithrananda et al., 2020) and its improved successor ChemBERTa-v2 (Ahmad et al., 2022) employ transformer-based language modeling to provide strong molecular embeddings from sequence data alone. Beyond purely textual inputs, SCAGE (Qiao et al., 2025) incorporates 3D conformational information through a self-conformation-aware graph transformer. UniMol-v1 (Zhou et al., 2023) further established 3D-informed pretraining by combining SE(3)-equivariant architectures with large-scale conformer data, while UniMol-v2 (Ji et al., 2024) scales this approach to the billion-parameter regime with substantially expanded datasets. More recently, MolFM (Luo et al., 2023a) introduced a multimodal formulation integrating molecular graphs, biomedical text, and knowledge-graph information into a unified embedding space, while GEM (Fang et al., 2022) captures molecular topology and geometry using a geometry-aware GNN and uses multiple self-supervised tasks to learn spatial knowledge from large-scale molecular structures.

