# OpenReview forum: "Learning the Neighborhood: Contrast-Free Multimodal Self-Supervised Molecular Graph Pretraining"
_ICML.cc/2026/Conference — ICML 2026 regular_

### Official Review · Reviewer_Qkre · 2026-02-26

**Soundness:** 3
**Presentation:** 4
**Significance:** 3
**Originality:** 3
**Overall Recommendation:** 4
**Confidence:** 3

**Summary:**

The paper proposes C-FREE (Contrast-Free Representation learning on Ego-nets), a self-supervised pretraining framework for molecular graphs that combines 2D molecular topology with 3D conformer ensembles. The method samples a node, extracts its k-hop ego-net as one view and uses its complement as the other view; it alternates which view is “context” vs. “target.” Both views are encoded with a hybrid GNN + 3D encoder + Transformer backbone (e.g., GINE for 2D and PaiNN/SchNet for 3D, then a Transformer), and training minimizes an L2 loss between a predictor output on the context embedding and a target embedding produced by an EMA-updated target encoder, in a BYOL/I-JEPA fashion (Fig. 1 and Sec. 3). The experiment results demonstrate that its strong performance on MoleculeNet datasets (linear probe and fine-tuning).

**Compliance With Llm Reviewing Policy:**

Affirmed.

**Final Justification:**

Thank you for the authors’ rebuttal. However, the choice of Ego-Net size still appears somewhat heuristic, and selecting an effective Ego-Net size cannot be justified solely through experimental results; a more comprehensive explanation or guideline is needed. In addition, a fairer experimental setup should be provided.

**Key Questions For Authors:**

1. How does performance change for $k$=1,2,5 and for different numbers of sampled anchor nodes per molecule? Do you observe failure modes where the complement view becomes too large/noisy or too small?
2. You state you omit positional encodings unlike GraphJEPA; can you provide intuition or an ablation showing positional encodings are not helpful (or harmful) for this multimodal token sequence?
3. Does C-Free perform better than other baselines that are trained on the same dataset?
4. Should any additional impact statements be included, given that C-Free could potentially be applied in drug discovery?

**Limitations:**

The paper includes an impact statement, but it does not meaningfully discuss limitations or potential negative impact. A concise limitations are: (i) view generation choice is somewhat heuristic, (2) fairness across baselines could be discussed more carefully.

**Strengths And Weaknesses:**

## Strengths
1. **Simple, well-motivated training objective.** The latent predictive JEPA/BYOL-style setup avoids negatives and avoids reconstructing discrete graph structure, which can be cumbersome for molecules. The “context = ego-net, target = complement (and vice versa)” is intuitive and easy to implement.
2. **Clarity and practicality.** The architecture and training recipe are described clearly (Sec. 3; App. A.4), and the approach avoids heavy preprocessing such as METIS clustering used in GraphJEPA.
3. **Strong empirical results across evaluation regimes.** The paper reports improvements under both frozen-backbone linear probing and full fine-tuning, with detailed comparisons against contrastive, non-contrastive, and multimodal baselines on MoleculeNet (Tables 1–2), plus QM9 (Table 3) and ZINC (Table 8).
## Weaknesses/concerns
1. **View generation choice is somewhat heuristic.** Fixed $k \in \{3,4\}$ ego-nets are motivated as “patch-like,” but it remains unclear how sensitive performance is to $k$, to sampling strategy (# ego-nets per molecule), and to complement construction for small graphs. More systematic sensitivity (beyond a brief discussion) would strengthen generality claims.
2. **fairness across baselines could be discussed more carefully.** The paper states that the model is pretrained on the GEOM dataset (~0.33M molecules), while some baseline models are trained on larger datasets. In my view, differences in the training data like scale and data distribution may significantly affect the experimental results. A clearer “matched-scale” comparison table and compute reporting (e.g., GPU-hours per baseline when reproduced vs. cited) would reduce ambiguity.

---

> ### Author Rebuttal · Authors · 2026-03-31
>
> We would like to thank Reviewer **Qkre** for their thoughtful review. We are glad that the reviewer highlighted the *“simplicity and strong motivation”* of our training objective, as well as *“the clarity and practicality”* of our approach. In what follows, we answer the Reviewer’s questions.
>
> **RW1a Analysis of Subgraphs for Different Ego-Net Sizes** To provide insight into the effect of ego-net size k, we compute structural features on subgraphs generated for k = 1, 2, 3, 4, 5 over 200 randomly sampled molecules from the GEOM dataset.
>
> The sampled molecules have an average size of **μ_atoms = 42** and an average connectivity of **μ_deg = 1.043**.
>
> The following table reports, for each value of k, the mean structural features aggregated over all context subgraphs: the number of atoms **(μ_atoms),** the number of bonds **(μ_bonds)**, the connectivity defined as the ratio of bonds to atoms **(μ_deg)**, and the coverage defined as the fraction of atoms in the context subgraph relative to the full molecule **(μ_cov).**
>
> |  | **μ_atoms** | **μ_bonds** | **μ_deg** | **μ_cov** |
> | --- | --- | --- | --- | --- |
> | k = 1 | 2.867 | 1.867 | 0.603 | 7.1% |
> | k = 2 | 7.133 | 6.133 | 0.849 | 17.6% |
> | k = 3 | 10.600 | 10.000 | 0.933 | 25.8% |
> | k = 4 | 15.600 | 15.533 | 0.989 | 38.%5 |
> | k = 5 | 21.067 | 21.200 | 1.003 | 52.4% |
>
> We observe that for k = 3, 4, the context subgraphs achieve a coverage of 26-52%, most likely  capturing meaningful local chemical environments, such as a functional group together with their immediate neighbors, while leaving enough information in the complementary subgraph to predict the missing signal. We will add this clarification in the appendix of the final version.
>
> **RW1b Effect of Ego-Net Sizes on Performance** We appreciate the reviewer's suggestion. In our pre-training setup, we introduce variation by sampling ego-nets of sizes k = 3, 4 rather than fixing a single value. Training a separate backbone for each individual k would require multiple long pre-training runs, so we opted for this mixed strategy instead. However, we agree that a more principled analysis of this parameter is required. Due to the substantial computational cost of pretraining, we are unable to provide this ablation within the rebuttal period. We commit to including a full ablation over ego-net sizes (k = 1, 2, 5) and number of sampled anchor nodes per molecule in the final version of the paper.
>
> ----------
>
> **RW2 Positional Encodings Omission** The omission of positional encodings is a deliberate design choice motivated by the need for permutation invariance. In our multimodal token sequence, tokens correspond to molecular modalities rather than sequentially ordered elements, and no canonical ordering exists across modalities.
>
> -------
>
> **RW3: Performance of C-FREE Pre-trained on Larger Datasets** We thank the reviewer for this suggestion. We agree that a fair comparison with the baselines requires pre-training C-FREE on the same larger datasets. We are currently running the pre-training experiments on the PCQM4Mv2, which is computationally demanding and requires significant time to complete. We commit to including the full results in the final version of the paper.
>
> -------
>
> **RW4 Additional Impact Statement** We have updated the manuscript to include an expanded discussion of the broader impact of C-FREE in the context of drug discovery.

---

> > ### Author Rebuttal · Reviewer_Qkre · 2026-04-02
> >
> > Thanks for your response. My concerns have been addressed.

---

> > > ### Author Response · Authors · 2026-04-03
> > >
> > > We again thank the reviewer for their constructive comments, which have helped us strengthen the paper. We hope we have managed to address all their concerns, and would kindly ask them to consider updating their score or confidence accordingly.

---

### Official Review · Reviewer_tUTQ · 2026-03-03

**Soundness:** 3
**Presentation:** 4
**Significance:** 3
**Originality:** 2
**Overall Recommendation:** 4
**Confidence:** 3

**Summary:**

This paper introduces a model and a training method for large-scale self-supervised graph representation learning.
The authors reach state-of-the-art results with a very simple training objective, as they simply minimize the discrepancy between two views of the same molecules. The two views (context and target) are constructed by considering the k-hop neighborhood of a node.
Importantly, the method avoids representation collapse with 2 simple components: 1) asymmetric encoders for "context" and "target" views, 2) Exponential Moving Average (EMA) parameter sharing between encoders. There is no need for contrastive learning.

**Compliance With Llm Reviewing Policy:**

Affirmed.

**Final Justification:**

The rebuttal was satisfactory.
Both the proposed model and training objective are well defined and sound. The theoretical results are interesting, and the authors report the downstream performances for a large number of tasks. The ablation studies are quite complete and convincing.

**Key Questions For Authors:**

1) Do you confirm that the transformer is permutation equivariant, i.e., that it uses no positional encoding? In that case, it feels like the separator tokens are unnecessary (since the tokens are treated as if they were unordered).
2) Can you provide more results about the effect of the decay rate $\tau$ on the performances? This seems like a very important mechanism to prevent representation collapse. Is it hard to tune?
3) Perhaps a naive question, but how are the 3D conformers obtained? Are they provided with the dataset or generated with some software? In that case, is it expensive and how stochastic is it, given that you use only 3 conformers to encode the molecule?
4) What about adding 1D SMILES as a third modality? It would fit perfectly in the proposed multimodal pipeline. I understand this may be out of the scope of the paper at this time, but this seems like an important discussion.

**Limitations:**

Yes

**Strengths And Weaknesses:**

(Soundness) Both the proposed model and training objective are well defined and sound. The theoretical results are correct and the authors report the downstream performances for a large number of tasks. The ablation studies demonstrate the importance of the asymmetric architecture. However, the ablation studies also reveal that the DeepSets aggregation does not significantly improve the performance, which questions the importance of Lemma 1. Currently, the paper does not provide any analysis of the robustness of the method w.r.t. the decay rate $\tau$ of the EMA, which seems key for preventing representation collapse. I am also confused about the requirement for separator tokens in the permutation-equivariant transformer (see questions).

(Presentation) Overall, the paper is well written. The tables are easy to read and Figure 1 illustrates the method clearly. I would say that the paper would benefit from introducing a few more notations. Defining $g$ to be the predictor, $\hat{y}$ the context, and $y$ the target, the training loss could be rewritten simply as
\begin{equation}
||g(f_\hat{\theta}(\hat{y})) - f_\theta(y) ||_2^2
\end{equation}
which, in my opinion, clarifies the method.

(Significance)The main technical strength of the paper is its ability to reach SOTA results with a simple method (both architecture, graph views, and training objective). This is valuable for the graph learning community. As often, the result on WL expressivity is weakly aligned with the reported empirical performances, which makes me question its relevance.

(Originality) As noted above, this work achieves state-of-the-art results with a simple approach that, in my opinion, should be considered a valuable novelty. Taken separately, all existing components of the model are classical. The paper also makes a first step toward comparing different views of a molecule (2D vs 3D), with a clear conclusion that the 3D view is much more expressive (to the point that it is not clear whether the model is truly multimodal or is discarding the 2D representation). This is an interesting direction, but it lacks discussion and comparison with "1D" SMILES representations.

---

> ### Author Rebuttal · Authors · 2026-03-31
>
> We would like to thank Reviewer **tUTQ** for their thoughtful review and positive feedback. We appreciate the reviewer’s comment that *“both the proposed model and training objective are well defined and sound”,* as well as their observation that *“the work constitutes a valuable novelty”*. We also thank the reviewer for the helpful suggestion regarding improvements to the notation, which we will incorporate in the revised manuscript. In what follows, we clarify the Reviewer’s concerns.
>
> **RW1 Necessity of separator tokens** The reviewer correctly observes that, given the permutation equivariance of the transformer, separator tokens should not strictly be necessary. Our design choice was primarily inspired by the approach of Manolache et al. [1], whose formulation we followed for consistency, given that the inclusion of these tokens is not computationally expensive. However, we agree that separator tokens are not strictly required but we hypothesize that they may serve as attention sinks [2], a role that could be helpful in our setting, since we do not use positional encodings. We will include an ablation study on separator tokens in the final version of the paper.
>
> -----------
>
> **RW2 Effect of Decay Rate** The reviewer rightly observes that while we hypothesize the decay rate is essential for preventing collapse, we do not tune this parameter. To address this, we include an ablation study examining the effect of the decay rate $τ$ = [$τ₀$, $τ_N$ ]. We follow the same setup as the modality ablation study and run pre-training on the GEOM dataset.
>
> We observe that the training loss is most sensitive to the choice of EMA decay rate. The pretraining loss starts at the same value for all 3 variants then evolves differently throughout training. A very high decay rate ($τ₀$ = 1) causes the loss to collapse abruptly after the first epoch and remain constant, while a lower decay rate ($τ₀$ = 0.8) leads to a noisier but faster decrease. The standard C-FREE setting ($τ₀$ = 0.995) yields slower but smoother convergence.
>
> Due to rebuttal time constraints, we were unable to run the pre-training to full convergence,  as this would require substantially more computational resources, but we include partial results in [Figure 1](https://figshare.com/s/266034a15f0f11fbea80). Since it is not clear how training loss maps to downstream performance, we cannot make any definitive claims about the effect of the decay rate. A full ablation study with corresponding probes will be included in the final version.
>
> -----------
>
> **RW3 Generation of the 3D conformers** For the generation of the conformers, we rely on the GEOM dataset, which provides pre-computed 3D structures. These are not generated on the fly using RDKit alone. In GEOM, initial molecular geometries are first generated with RDKit, then optimized using GFN2-xTB [3], and finally sampled with CREST [4].
>
> ------
>
> **RW4 Addition of 1D Modality** We chose to exclude SMILES for both practical and conceptual reasons. Practically, our self-supervised objective masks k-EgoNet subgraphs defined by graph topology, and since SMILES is a serialization of the molecular graph, there is no clean correspondence between a masked subgraph and a contiguous region of the SMILES string. Designing a masking strategy that occludes the same chemical information across modalities without leaking structural information to the other views would therefore be non-trivial. Conceptually, SMILES and 2D molecular graphs encode largely redundant information, consistent with findings in the supervised multimodal setting [1], where the 1D modality contributed least.We thank the reviewer for raising this point and will include a discussion of SMILES in the final revision of the paper.
>
> [1] Manolache, Andrei, Dragos Tantaru, and Mathias Niepert. "Molmix: a simple yet effective baseline for multimodal molecular representation learning." arXiv preprint arXiv:2410.07981 (2024).
>
> [2] Xiao, Guangxuan, et al. "Efficient streaming language models with attention sinks, 2024." URL https://arxiv. org/abs/2309.17453 (2023): 1.
>
> [3]: Bannwarth et al, “GFN2-xTB - An Accurate and Broadly Parametrized Self-Consistent Tight-Binding Quantum Chemical Method with Multipole Electrostatics and Density-Dependent Dispersion Contributions”, Journal of Chemical Theory and Computation, 2019
>
> [4]: Pracht et al, “CREST - A program for the exploration of low-energy molecular chemical space”, The Journal of Chemical Physics, 2024

---

> > ### Author Rebuttal · Reviewer_tUTQ · 2026-04-02
> >
> > Thank you for your answer. RW1, RW3 and RW4 are fully addressed in your answer. Please add these discussions to the paper.
> >
> > Regarding RW2, I understand the limitations due to the rebuttal, but it really seems that the performances of the method are directly dependent on the choice of the scheduling for $\tau$.
> > A detailed ablation study would strengthen the paper considerably and increase its practical relevance.

---

> > > ### Author Response · Authors · 2026-04-03
> > >
> > > We again thank the reviewer for their constructive comments, which have helped us strengthen the paper.
> > >
> > > We also provide intermediate results using the same EMA instances from RW2 in [Figure 2](https://figshare.com/s/82eb13a615cac8132a35), with probes run on the Kraken L target after every 5 epochs. When using the High EMA, probe loss collapses immediately and fails to improve throughout pre-training. The Low EMA yields a noisy loss, while the C-FREE variant produces the most stable behavior.These findings are preliminary; the run has not yet converged, so we will need to wait until completion to draw stronger conclusions. We will also run more rigorous experiments to confirm this behavior for the final version of the paper.
> > >
> > > We hope we have managed to address all their concerns, and would kindly ask them to consider updating their score or confidence accordingly.

---

### Official Review · Reviewer_U7sP · 2026-03-11

**Soundness:** 4
**Presentation:** 2
**Significance:** 4
**Originality:** 3
**Overall Recommendation:** 5
**Confidence:** 4

**Summary:**

The manuscript presents a self-supervised representation learning method using multimodal input: 2D structure and multiple 3D conformations of compounds. The training optimizes performance on the self-supervision task of predicting the embedding of a subgraph using its complementary subgraph. Empirical results with multiple dataset and fine-tuning strategies demonstrate state of the art performance.

**Compliance With Llm Reviewing Policy:**

Affirmed.

**Final Justification:**

The problem of molecular representation learning is a relevant problem for multiple scientific and industrial applications.
The method is well motivated, sufficient details are given for reproducibility, it can incorporate information from multiple 3D conformations .The empirical evaluation is thorough, and show SOTA results. During the rebuttal the authors addressed my issue with the error bars and "significance" levels.

**Key Questions For Authors:**

- Define EgoNet and k-EgoNets rigorously! While there is a drawing on Figure 6 (Labeled as Table 6), there are some ambiguities in this figure: While in the text it is clearly written that the subgraphs are disjoint, or complementary, on the image two carbons of the oxolane ring (position 3 and 4) is included both in the context and the target, while the other two carbon (position 2 and 5) only in the context. I also tried to check the cited work (Bevilacqua et al.,2022), but also did not see a precise defintion of EgoNets (i did not read the entire paper of Bevilacqua et al., but scanned it)
- How is your model's performance with tasks where chirality is relevant? GEOM and other QM9 derived datasets have been criticized as they not provide enough signal for chirality. This is likely because these moleculse have to be limited in size for DFT calculations be feasable. (eg.: L. Pattanaik, O.-E. Ganea, I. Coley, K. F. Jensen,
W. H. Green, and C. W. Coley, “Message Passing Networks for Molecules with Tetrahedral Chirality,” (2020), arXiv:2012.00094 )
- You write that the predictor only takes the embedding of the CLS token [L207, left col.], than you write it is a transformer. Other places you write that everything is taken from the context encoder, and the predictor network give out CLS token embedding. (L181 right col.) It is a bit confusing, can you clarify?
- What is the reasoning behind your choice to use the target encoder and not the context encoder in inference time?
- Please use a coloring/bolding scheme in the tables where you select all indistinguishable results. Quick and dirty way is to select those where the error interval overlaps, this approximately corresponds to 0.05 significance level in t-test if you have 5 repreats. Better even to do some stat. tests. How many repeats you have?

**Limitations:**

The authors mention possible limitations, but it would be valuable to see a comment of chirality, if there is a limitation there, see Q2.

**Strengths And Weaknesses:**

Strengthes:
- The problem of molecular representation learning is a relevant problem for multiple scientific and industrial applications.
- Incorporating information from multiple 3D conformations is an active area of research.
- The architecture and training method is well motivated, and given in details.
- The empirical evaluation is thorough, and show SOTA results.

Weaknesses:
- Ego-net are not well defined, the given figure is ambiguous, see questions Q1
- Some part of the architecture description is confusing, see Q3.
- At some case the statistical evaluation of results need more rigor. In Table 1 for example Tox21 column, $76.1\pm 0.2$ and $76.7 \pm 0.6$ is likely indistinguishable except if you have well over 10 repeats. Same problem even more pronounced in cols SIDER and BACE. Some similar issue also visible in Table 2.  See Q5.

---

> ### Author Rebuttal · Authors · 2026-03-31
>
> We would like to thank Reviewer **U7sP** for their thoughtful review. We appreciate the reviewer’s recognition that our *“architecture and training method are well motivated”* and that *“the empirical evaluation is thorough.”* In the following, we address and clarify the reviewer’s concerns.
>
> **RW1 Clarification of k-EgoNet definition** We thank the reviewer for pointing out the missing formal definition of k-EgoNet subgraphs. Following ESAN, we define the k-ego-net of a node as its k-hop neighbourhood with the induced connectivity. Building on this, we define complementary subgraphs as follows: the context consists of the k-EgoNet of an anchor node, while the target comprises the remaining edges not included in the context. To guarantee full edge coverage, edges on the boundary between the two subgraphs are assigned exclusively to one subgraph; boundary nodes, however, are duplicated and appear in both, ensuring the two subgraphs are edge-disjoint.
>
> The main figure in the manuscript illustrates this with a 2-EgoNet centered on the circled carbon atom. All edges incident to this anchor node are included in the context. The two carbon atoms of the oxolane ring fall within its 2-hop neighbourhood and are therefore also part of the context, and their connecting edge as well. Since these atoms additionally connect to nodes in the target subgraph, they are duplicated and included in the target as well. We will update the manuscript to incorporate this formal definition.
>
> -------------
>
> **RW2 Performance on Chirality-Sensitive Tasks** We thank the reviewer for the insightful comment regarding chirality and its potential incompatibility with our current encoders. We fully agree that both GEOM and QM9 datasets, as well as our encoders (SchNet and PaiNN) provide limited signal for chirality. For reference, preliminary results on a subset of the PubChem3D dataset and the R/S classification task from [1] are: SchNet: 53.3%, PaiNN: 52%, and C-FREE: 52%.
>
> We would like to highlight that our framework is fully encoder-agnostic, which makes it possible to integrate encoders with specific biases, such as chirality-aware models. While this is beyond the scope of the current work, exploring such encoders or combining them as  a form of different modalities represents a very promising direction for future research.
>
> We greatly appreciate this thoughtful suggestion and will discuss it as a limitation in our paper. Conducting a full evaluation would ideally involve SSL pretraining on a dataset with chirality-specific information, which we consider a compelling direction for future experiments.
>
> --------------
>
> **RW3 Clarification of the Pooling Method in the Predictor** We thank the reviewer for identifying this inconsistency, and apologize for the oversight. To clarify: rather than applying explicit pooling, we pass the full sequence through the predictor and rely on the CLS token to aggregate global information. This CLS token effectively acts as the pooled representation, and is what gets compared against the CLS token produced by the target encoder. We have updated the manuscript accordingly to correct this error.
>
> ------------
>
> **RW4 Choice of Downstream Encoder** We use the target encoder at inference time, though in practice the choice is inconsequential: by the end of training, the context and target encoders converge to the exact same weights due to the EMA decay parameter being set to τ = 1.
>
> -------------
>
> **RW5 Improved Coloring in the Results Tables** The reviewer rightly points out that our highlighting of best results does not account for overlapping confidence intervals. For Tables 1 and 2, we follow Wang et al. [2] and Liu et al. [3] in running each model with 3 random scaffold seeds to ensure fair comparison. Given this small number of runs, a formal statistical test would not be appropriate.
>
> To improve rigor, we will update the tables to reflect statistical indistinguishability: any result whose interval [mean − std, mean + std] overlaps with the best result's interval [best_mean − std, best_mean + best_std] is now highlighted with the same color as the best result. Using this strategy, however, does not change the overall results and C-FREE remains the best scoring method.
>
> [1] Adams, Keir, Lagnajit Pattanaik, and Connor W. Coley. "Learning 3d representations of molecular chirality with invariance to bond rotations." *arXiv preprint arXiv:2110.04383* (2021).
>
> [2] Wang, Hanchen, et al. "Evaluating self-supervised learning for molecular graph embeddings." *Advances in Neural Information Processing Systems* 36 (2023): 68028-68060.
>
> [3] Liu, Shengchao, et al. "A group symmetric stochastic differential equation model for molecule multi-modal pretraining." *International Conference on Machine Learning*. PMLR, 2023

---

> > ### Author Rebuttal · Reviewer_U7sP · 2026-04-03
> >
> > Thank you for your detailed answers!

---

> > > ### Author Response · Authors · 2026-04-03
> > >
> > > We again thank the reviewer for their constructive comments, which have helped us strengthen the paper. We hope we have managed to address all their concerns, and would kindly ask them to consider updating their score or confidence accordingly.

---

### Official Review · Reviewer_u8i7 · 2026-03-12

**Soundness:** 3
**Presentation:** 3
**Significance:** 3
**Originality:** 3
**Overall Recommendation:** 5
**Confidence:** 4

**Summary:**

The authors propose a new self-supervised pretraining method for molecular representation learning that incorporates both 2D and 3D information, called Contrast-Free Representation learning on Ego-nets (C-FREE).

An k-ego-net is a local subgraph centered on a specific node consisting of that node and all atoms reachable within k bond hops. In C-FREE, ego-nets serve as the basis for subgraph partitioning: a random atom is sampled from a molecule, its k-hop ego-net is extracted as the context subgraph, and the remaining atoms constitute the complementary target subgraph. The intuition is that many local chemical environments are functionally similar, so predicting one neighborhood from another forces the model to learn chemically meaningful representations.

The core training objective is inspired by the Joint Embedding Predictive Architecture (JEPA). Rather than reconstructing masked atoms or bonds directly, the model is trained to predict the latent representation of the target subgraph from the latent representation of the context subgraph. This keeps all supervision in embedding space and avoids the low-level reconstruction bias and noise of predicting in the input space. The architecture comprises three components: a context encoder, a target encoder, and a predictor.

Both encoders share the same hybrid architecture:
- a message-passing GNN (GINE) for 2D topology,
- an equivariant network (PaiNN or SchNet) for 3D conformer geometry,
- and a Transformer that fuses the two modalities via a learnable multimodal sequence.

Only the context encoder is trained by backpropagation; the target encoder's weights are an exponential moving average (EMA) of the context encoder's parameters, updated without gradients. This asymmetry between a trainable predictor and a slowly-moving target encoder is what prevents representational collapse. The predictor, implemented as a Transformer, maps the context embedding to the predicted target embedding, and training minimises the mean squared error between this prediction and the actual target encoder output.

The authors claim that this framework integrates both topological and geometric molecular information without requiring negative examples (typical of contrastive learning), explicit positional encodings (used by GraphJEPA and I-JEPA), or costly graph clustering procedures such as METIS. The result is a conceptually simpler pipeline than prior art while remaining theoretically grounded: the authors show that C-FREE with a DeepSets aggregation head is at least as expressive as ESAN, hence strictly more expressive than the 1-WL Weisfeiler-Leman test.

On the empirical side, the paper reports improvements on the MoleculeNet classification benchmarks over both contrastive and non-contrastive baselines, with effective transfer to new chemical domains including conformer-resolved regression on the Kraken dataset and label-efficient learning on the larger Drugs-75K benchmark. Ablations confirm that the predictor is critical to prevent collapse and that 2D and 3D modalities provide complementary signals, with 3D contributing the larger share of downstream performance gains.

**Compliance With Llm Reviewing Policy:**

Affirmed.

**Final Justification:**

W1 - QM9 evaluation

The authors provide both the requested full-backbone results (Table 1 in the rebuttal) and the PaiNN-from-scratch baseline (Table 2). The baseline confirms that C-FREE's SSL objective contributes meaningfully beyond PaiNN's inductive bias on 5 of 6 targets, partially resolving the concern. The architectural mismatch with UniMol2, which the review flagged as an interpretation risk, is also now better contextualized by the 3D-only alignment rationale, even if the magnitude of gains over UniMol2 on some QM9 targets remains striking and is not fully explained.

Q3 - data overlap in the label efficiency experiment

The authors confirm that Drugs-75K molecules appear in the pretraining corpus at the level of 2D topology, while clarifying that the downstream conformers are generated via a different pipeline (Auto3D/AIMNet-NSE vs. CREST/GFN2-xTB used in GEOM). The authors answer is that the pretraining objective is fully label-free and the geometric inputs differ - it's a matter of debate whether that could be considered leakage or not. I would personally exclude from the pre-training set all smiles that overlap with the test split - label free or otherwise. The authors commit to adding an explicit note in the revision.

L1 - SIDER underperformance

The explanation, that dummy coordinates are used when conformer generation fails for large and complex molecules, explains the difference. Adding this clarification to the paper is a resolves the limitation.

Overall, C-FREE presents a well-motivated simplification of the molecular SSL pipeline with solid empirical support across multiple benchmarks. I maintain my recommendation of Accept.

**Key Questions For Authors:**

Q1. The QM9 evaluation departs from the standard multimodal setup (GINE + PaiNN/SchNet + Transformer) used elsewhere, restricting C-FREE to its 3D encoder only. Could the authors clarify why, and provide results with the full backbone?

Q2. Could the authors provide a randomly initialized PaiNN baseline on QM9, analogous to the RND baselines in Tables 4 and 5? Given that pretraining yields only modest gains over random initialization in those experiments (~1.5–7%), this baseline would isolate the contribution of the SSL objective from PaiNN's inductive biases.

Q3. The label efficiency experiment (Table 4) compares a pretrained backbone (FFT) against random initialization (RND) on Drugs-75K. Drugs-75K is described as a subset of GEOM-Drugs, which is part of the pretraining corpus. Could the authors confirm whether Drugs-75K molecules appear in the pretraining split, and if so, report the degree of overlap?

**Limitations:**

L1. No discussion of failure modes, e.g. SIDER underperformance is never explained. UniMol gives reason for it (3D conformer generation failing for complex natural products and peptides).

**Strengths And Weaknesses:**

## Strengths

**S1. Simplicity vs. performance.**

A recurring pattern in molecular SSL is accumulating complexity e.g. contrastive objectives, positional encodings, expensive clustering procedures like METIS. C-FREE removes all of these, replacing them with ego-net partitioning and a JEPA-style predictive objective, and the simplification does not come at a performance cost.

C-FREE matches or outperforms more complex baselines on 6 of 8 MoleculeNet tasks under linear probing, and ranks first overall on full fine-tuning despite being pretrained on 10–200x fewer molecules than the strongest competitors.

**S2. Theoretical grounding with empirical validation.**

Lemma 1 establishes that with a DeepSets aggregation head, the model is at least as expressive as ESAN and strictly more expressive than 1-WL, and Table 7 empirically validates this on the EXP dataset, where 1-WL GNNs cannot exceed chance. The authors are also intellectually honest about the limits of this result, acknowledging directly that 1-WL already separates most molecular graphs in practice (Pellizzoni et al., 2025), so the practical impact is limited.

## Weaknesses

**W1. QM9 results**

C-FREE is a much smaller model than UniMol2, both in terms of the training examples (330k vs. 800M) and in terms of the number of parameters (9.1M vs 1.1B), however reports astonishing gains of 2-4x on several targets from QM9. These results are very impressive, but require further explanation.

For this benchmark the authors depart from their standard evaluation setup, restricting C-FREE to its 3D encoder only (PaiNN/SchNet). PaiNN in particular is an SE(3)-equivariant message-passing network designed specifically for quantum chemical property prediction, operating directly on interatomic distances and angles. UniMol2, by contrast, uses a general-purpose Transformer architecture. This architectural mismatch makes it difficult to attribute the observed gains to C-FREE's pretraining objective rather than to the inductive biases of the backbone.

Fine-tuning a randomly initialized PaiNN on QM9 without any SSL pretraining could isolate the effect of pretraining and the architecture.

The authors run exactly this kind of comparison in analogous experiments in Table 4 and Table 5. The results there show modest gains over random initialization, ~1.5-7% on Kraken, and ~5% on Drugs-75k.

---

> ### Author Rebuttal · Authors · 2026-03-31
>
> We would like to thank Reviewer **u8i7** for their thoughtful review. We are pleased that the reviewer recognized and appreciated several complementary aspects of our contribution, in particular that *"the simplification does not come at a performance cost"* and that the work provides *"theoretical grounding with empirical validation all while acknowledging the limits of the theoretical result"*. In what follows, we answer the Reviewer’s questions.
>
> **RW1 Justification for the QM9 Experimental Setup** Reviewer u8i7 correctly notes that our QM9 experiment is restricted to the 3D+Transformer configuration rather than the full multi-modal setup. This design choice is to ensure a fair comparison with the provided baselines (GEM and UniMol).
>
> Concretely, GEM employs a geometry-augmented graph neural network that integrates bond lengths and angles through coupled atom-bond and bond-angle graphs, while UniMol’s 3D-aware Transformer operates directly on molecular conformations via distance- and position-informed attention. Thus both methods rely solely on the geometric structure of the graph. Aligning our configuration with this 3D-only regime avoids giving our method an unfair advantage from richer inputs and isolates the contribution of our architectural choices within the same information setting.
>
> To provide the full picture, results using the full backbone are reported in Table 1.
>
> **Table 1: Full fine-tuning using Mutli-modal setup on QM9**
>
> | Method | Mu | Alpha | HOMO/LUMO/GAP | R2 | Cv | ZPVE |
> | --- | --- | --- | --- | --- | --- | --- |
> | C-FREE 3D | 0.064 | 0.115 | **0.00401** | **1.155** | 0.056 | 0.0061 |
> | C-FREE MM | **0.054** | **0.103** | 0.00435 | 1.243 | **0.044** | **0.0053** |
>
> While the 3D variant shows a slight advantage on R2 and HOMO/LUMO/GAP, the multi-modal variant of C-FREE shows overall the best performance. We will also report standard deviations over multiple runs in the final version of the paper.
>
> -------
>
> **RW2 Isolating PaiNN's Inductive Bias** Reviewer u8i7 suggests training a PaiNN model from scratch to disentangle the contribution of the SSL objective from PaiNN's inherent inductive bias. We address this by providing the requested baseline in Table 2, using the same experimental setup as the QM9 experiment.
>
> **Table 2: Comparison of C-FREE vs PaiNN performance on QM9**
>
> | Method | Mu | Alpha | HOMO/LUMO/GAP | R2 | Cv | ZPVE |
> | --- | --- | --- | --- | --- | --- | --- |
> | C-FREE 3D | 0.064 | **0.115** | **0.00401** | **1.155** | **0.056** | **0.0061** |
> | PaiNN | **0.045** | 0.145 | 0.00809 | 1.226 | 0.066 | 0.0112 |
>
> We observe that C-FREE performs better on 5 out of the 6 targets, while PaiNN shows a slight advantage on Mu, suggesting that C-FREE's design choices contribute meaningfully beyond PaiNN's inherent inductive bias. We will also report standard deviations over multiple runs in the final version of the paper.
>
> ------------
>
> **RW3 Data Overlap in the Label Efficiency Experiment** We confirm that there is an overlap between Drugs-75K and the pretraining set, as Drugs-75K is a subset of GEOM-Drugs. However, this overlap concerns only the shared 2D molecular topology, while the conformers are generated through different pipelines. Specifically, GEOM-Drugs conformers are produced using CREST at the GFN2-xTB level of theory, whereas the Marcel conformers are generated and optimized independently using Auto3D, with AIMNet-NSE employed to compute DFT-based reactivity descriptors. Consequently, the conformers used in the downstream task are not identical to those seen during pretraining.
>
> Additionally, although the underlying molecular 2D structures may appear during pretraining, the objective is fully self-supervised and label-free, meaning the model is never exposed to downstream task labels at this stage. We agree that clarifying this distinction is important and will add an explicit note in the next revision.
>
> -----------
>
> **RW4 Explaining the Underperformance on SIDER** We thank Reviewer u8i7 for raising this point. As also noted in UniMol, conformer generation fails for the larger molecules present in SIDER; we circumvent this by using dummy coordinates for this dataset. This limitation directly accounts for the observed underperformance. We have added an explicit clarification to the next revision.

---

> > ### Author Rebuttal · Reviewer_u8i7 · 2026-04-02
> >
> > Thank you for the clarification. My questions have been addressed.

---

> > > ### Author Response · Authors · 2026-04-03
> > >
> > > We are grateful for the reviewer's positive assessment of our work. We hope the clarifications provided are satisfactory, and we remain available to address any remaining questions.

---

### Decision · Program_Chairs · 2026-04-30

**Decision:**

Accept (regular)

**Comment:**

The paper presents C-FREE, a self-supervised molecular pretraining method that combines 2D graph structure with 3D conformers, without relying on a purely contrastive objective. It works by predicting subgraph embeddings of molecules from their complementary ego-neighborhoods in latent space, leveraging asymmetric encoders and exponential moving average (EMA) target weights instead of negatives or explicit reconstruction.

The paper was broadly well received. Reviewers praised the simplicity of the objective, the multimodal 2D-3D integration, and the strong benchmark results. Concerns mainly revolved around formal definitions, chirality limitations, ego-net size and EMA ablations, and fairness against larger-data baselines. The rebuttal successfully addressed the majority of the concerns (only minor remain, all addressable in post-decision), and reviewers explicitly marked their concerns as resolved.

I recommend acceptance.